# A CRISPRi screen in *E. coli* reveals sequence-specific toxicity of dCas9

Lun Cui [1], Antoine Vigouroux[1], François Rousset [1,2], Hugo Varet[3], Varun Khanna[3] & David Bikard [1]

High-throughput CRISPR-Cas9 screens have recently emerged as powerful tools to decipher gene functions and genetic interactions. Here we use a genome-wide library of guide RNAs to direct the catalytically dead Cas9 (dCas9) to block gene transcription in *Escherichia coli*. Using a machine-learning approach, we reveal that guide RNAs sharing specific 5-nucleotide seed sequences can produce strong fitness defects or even kill *E. coli* regardless of the other 15 nucleotides of guide sequence. This effect occurs at high dCas9 concentrations and can be alleviated by tuning the expression of dCas9 while maintaining strong on-target repression. Our results also highlight the fact that off-targets with as little as nine nucleotides of homology to the guide RNA can strongly block gene expression. Altogether this study provides important design rules to safely use dCas9 in *E. coli*.

---

[1] Synthetic Biology Group, Department of Microbiology, Institut Pasteur, Paris 75015, France. [2] Sorbonne Université, Collège Doctoral, Paris F-75005, France. [3] Hub Bioinformatique et Biostatistique, Institut Pasteur - C3BI, USR 3756 IP CNRS, Paris 75015, France. Correspondence and requests for materials should be addressed to D.B. (email: david.bikard@pasteur.fr)

Over the past few years, tools derived from the bacterial immune system known as Clustered Regularly Interspaced Short Palindromic Repeats (CRISPR) and the associated *cas* genes have led to many breakthroughs in genome editing and the control of gene expression[1]. In particular the Cas9 protein has proven to be a very versatile RNA-guided nuclease[2]. Target search goes as follows: Cas9 first scans DNA for the presence of the protospacer adjacent motif (PAM), a small sequence pattern of 2–8 nucleotides (nt)[3,4]. It then initiates the formation of an R-loop by pairing the guide RNA to the target starting from the PAM-proximal region, also known as the seed sequence[5]. The catalytic dead variant of the RNA-guided Cas9 nuclease, known as dCas9, binds to target positions without cleaving DNA[6]. In bacteria, directing dCas9 to bind a promoter region blocks the initiation of transcription, while binding the non-template strand downstream of a promoter efficiently stops elongation of transcription[6,7]. This provides a convenient method to silence genes that has already been used to investigate the role of essential genes in *Bacillus subtilis* and *Streptococcus pneumoniae* via high-throughput screens[8,9].

The action of Cas9 at off-target positions is a major concern for genome-editing applications[10–12], as it could lead to undesired mutations. While extensive binding beyond the seed sequence is required for a conformational shift in Cas9 to occur leading to DNA cleavage[13,14], chromatin immunoprecipitation sequencing experiments have revealed that Cas9 can bind to target positions with as little as 5 nt of homology between the seed region of the guide RNA and the target[15,16], possibly binding hundreds of positions in genomes. These results are also consistent with in vitro assays showing that dCas9 binding to its target remains unaffected by up to 12 mismatches in the PAM-distal region[17], as well as evidence that DNA binding guided by as little as 10 bases can be sufficient for dCas9 to have an effect on transcription in *Escherichia coli*[7]. These results are, however, in sharp contrast to what was reported in a study of dCas9-mediated repression or activation in human cells, where activity was highly sensitive to mismatches[18]. While substantial work has already been conducted to characterize the off-target activity of Cas9 in eukaryotes for genome-editing applications, comparatively little has been done for dCas9 in general and in bacteria in particular.

In this study, we performed a genome-wide pooled dCas9 knockdown screen in *E. coli* with the initial purpose of uncovering the properties and design rules of such screens. This screen confirmed previously reported properties of dCas9 repression in bacteria but revealed the presence of many guides producing unexpectedly strong fitness defects. A combination of machine learning and experimental approaches enabled us to attribute the effect of these guides to two main causes: (i) off-target binding positions that can block the expression of essential or fitness genes with as little as 9 nt of identity in the seed sequence, and (ii) an unexplained sequence-specific toxicity effect that is determined by the 5 PAM-proximal bases and that we refer to as the "bad-seed" effect.

## Results

**Effect of dCas9-binding position and orientation**. We designed a library of ~92,000 unique guide RNAs targeting random positions along the genome of *E. coli* MG1655, with the simple requirement of a "NGG" PAM. The library contains an average of 19 targets per gene. A pool of guide RNAs obtained through on-chip oligo synthesis was cloned under the control of a constitutive promoter on plasmid psgRNA and electroporated in strain LC-E18 carrying the dCas9 gene under the control of a Ptet promoter in the chromosome (Supplementary Fig. 1). The pooled library of cells was then grown in rich medium over 17

generations with anhydrotetracycline (aTc). The effect of each guide on the cell fitness can be measured as the fold change in abundance (log2FC) of the guide RNA in the library during the course of the experiment, as measured through deep sequencing of the library.

In order to investigate the properties of dCas9 repression in *E. coli*, we can analyze the effect of guides targeting essential genes. We expect guides that efficiently block the expression of these genes to be depleted from the library. Previous reports suggested that dCas9 efficiently blocks transcription elongation only when binding the coding strand (non-template strand)[6,7]. As expected guide RNAs targeting essential genes have on average a strong fitness effect when they bind to the coding strand and no fitness effect when they bind to the template strand (Fig. 1a). On the other hand, dCas9 binding in both orientations was reported to efficiently block the initiation of transcription. We analyzed the effect of guides binding the promoter region of a subset of 64 essential genes whose promoter is well defined (Supplementary Data 1). Our results also corroborate these findings (Fig. 1b).

Since dCas9 blocks transcription, we expect that targeting a gene in an operon will also silence all the downstream genes. Guides targeting non-essential genes upstream of essential genes in operons indeed showed a strong fitness defect (see the examples of *cydDC* and *ycaR-kdsB* operons in Fig. 1c). A reverse polar effect was reported in *B. subtilis* where targeting downstream of a gene was seen to block the expression of the upstream gene likely through destabilization of the interrupted transcript[8]. In our screen, we can find many examples where targeting a non-essential gene downstream of an essential gene does not have an impact on the cell fitness (see the examples of *rpoZ-spoT-trmH-recG* and *psd-mscM* operons in Fig. 1d). Opposite examples where targeting the downstream non-essential gene does have an effect can also be found, but in these cases the non-essential gene is typically known to be required for normal growth or is itself followed by another essential gene. These observations suggest that translation can still efficiently occur from mRNAs interrupted by dCas9 in *E. coli*. We did nonetheless observe that guides targeting within ~100 nt after the stop codon of an essential gene sometimes produced a fitness defect. This can, for instance, be seen for a guide in Fig. 1c targeting just after *kdsB*. To study this in a more systematic way, we compiled a list of essential or fitness genes that are not followed by another essential or fitness gene (Supplementary Data 2). Guides targeting within 100 nt of the end of these genes on the coding strand indeed produce a weak but significant fitness defect (Fig. 1e, single sample *t*-test comparison to the mean log2FC of guides targeting the template strand of genes, *p*-value < $10^{-4}$). On the other hand, guides targeting 100–200 nt after the end of these genes did not show a significant effect. A reverse polar effect of dCas9 on the expression of upstream genes thus does seem to exist in *E. coli*, but it is likely short range and weak.

Previous reports suggested that dCas9-mediated repression is negatively correlated with the distance from the beginning of the gene[6]. The same list of genes was used to look at the effect of the relative distance along the gene (Fig. 1f). No effect could be seen: dCas9 efficiency does not seem to correlate with the position inside the gene.

**Guides producing unexpected fitness defects**. Surprisingly, we observed a high variability of fitness effects between guide RNAs targeting nearby positions in the same orientation (Fig. 2a). These effects are reproducible between three independent experiments, suggesting that they are not the product of experimental noise but a real biological effect. In particular guides, binding to the template strand of non-essential genes are not expected to be

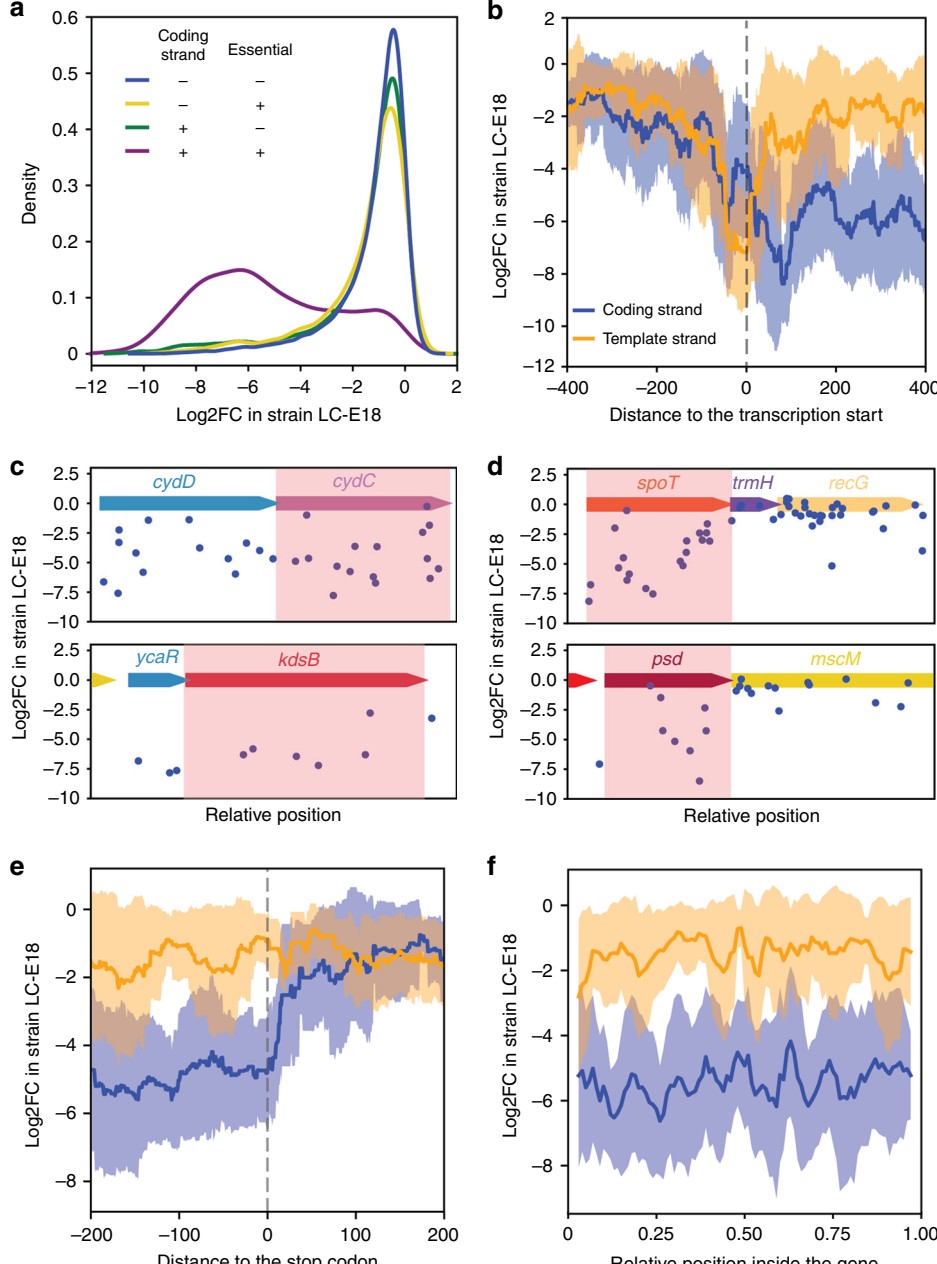

**Fig. 1** Effect of dCas9 binding position and orientation. **a** Distribution of the fitness effect of guide RNAs in our library depending on target gene essentiality and target strand. **b** Rolling average of the fitness effect produced by guides targeting the promoter of essential genes in both orientations (rolling window size of 50 bp). **c** Examples of polar effect seen in the *cydDC* and *ycaR-kdsB* operons. Gene *cydC* and *kdsB* highlighted in red are essential but not gene *cydD* and *ycaR*. Guides binding to the coding strand are shown as blue dots. **d** Example of operons containing an essential gene followed by a non-essential gene. Targeting the downstream non-essential gene usually does not produce a fitness defect. **e** Rolling average of the fitness effect produced by guides targeting the end of essential genes (rolling window size of 50 bp). **f** Rolling average of the fitness effect produced by guides along the length of essential genes. Gene start is 0 and gene end is 1 (rolling window size is 5% of the gene length). In all rolling average plots, the shaded area represents the standard deviation

depleted from the library, but in fact 7% of these guides (2499/ 36,111) produce a strong fitness defect (log2FC < −3.5, see Methods), accounting for 34% of all guides producing a strong fitness defect (Supplementary Table 1).

We arbitrarily decided to further investigate two such guides targeting *lpoB* (T-lpoB) and *hisI* (T-hisI). As a control, for each gene we also analyzed the effect of a nearby guide RNA targeting the coding strand (C-lpoB, C-hisI). Cells carrying T-lpoB or T-hisI only show a small reduction of the target gene transcription but show a markedly reduced plating efficiency

when dCas9 is induced, consistently with the screen results (Fig. 2b, c). On the contrary, the C-lpoB and C-hisI guides strongly block the transcription of their target gene but show no defect in plating efficiency. Since a moderate repression of a non-essential gene is very unlikely to cause the death of *E. coli*, we hypothesized that this phenotype results from off-target activity. When looking for putative off-target positions for the T-hisI sgRNA, we identified an 11 base pair (bp) perfect match between the seed region of the guide RNA and a putative off-target in the promoter of the *dnaK-dnaJ* operon. We could measure that the

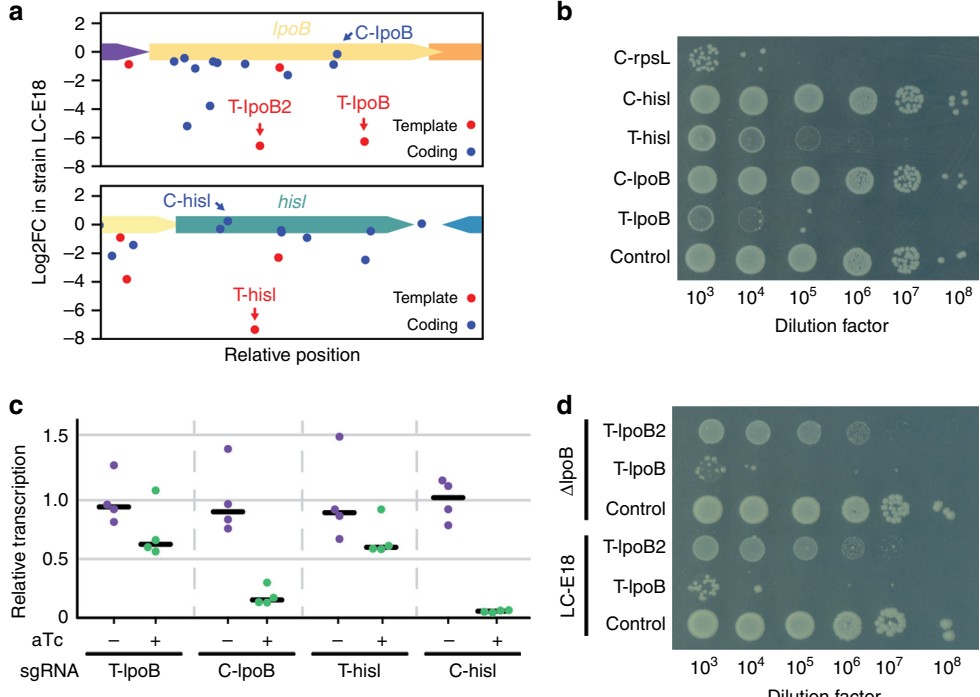

**Fig. 2** Analysis of guides producing unexpected fitness defects. **a** Fitness effect of guides targeting the non-essential *lpoB* and *hisI* gene. Guides T-lpoB, T-lpoB2, C-lpoB, T-hisI, and C-hisI are highlighted. **b** The highlighted guides were cloned in plasmid psgRNA and introduced in strain LC-E18 carrying dCas9 under the control of a Ptet promoter in the chromosome. Cells were grown overnight and plated on LB agar with 1 nM of aTc. Representative figure from *p* = 4 independent experiments. **c** Expression levels of *lpoB* and *hisI* when repressed by the T-lpoB, C-lpoB, T-hisI, and C-hisI guide RNAs, as measured by RT-qPCR. Points show biological replicates (*n* = 4), the black bar is the median. **d** Effect of the T-lpoB, T-lpoB2, and control guides on the plating efficiency of LC-E18 and LC-E18ΔlpoB. Cells were grown overnight and plated on LB agar with 1 nM of aTc. Representative figure from *n* = 4 independent experiments

T-hisI guide RNA produces a three-fold repression of the *dnaK-dnaJ* operon. While *dnaK* or *dnaJ* are not essential genes, our screen results show that guides blocking the expression of this operon consistently produce a strong fitness defect (Supplementary Fig. 2). This off-target position thus likely explains the fitness defect produced by T-hisI. To investigate the preponderance of guides with such off-targets, we looked for off-targets that displayed a perfect identity of 9 nt or more between the seed sequence and regions where a strong fitness defect was consistently observed (i.e. essential or fitness genes). We found such off-targets for 24% (600/2499) of the guides that produced an unexpected fitness defect. As a control, we also looked at the proportion of guides with such off-targets among guides that target the same genes in the same orientation but produce no fitness defects. This occurs for 10.7% (3609/33,612) of these guides, giving a measure of the false-positive discovery rate. Guides that produce unexpected fitness defects are thus significantly more likely to have an off-target blocking the expression of a fitness or essential gene (Fisher exact test *p*-value < 0.001). This enables to provide a conservative estimate that the fitness defect produced by 13% of these guides is due to their off-target activity.

When doing this analysis, it became evident that no obvious off-target positions could be identified for a large majority of guide RNAs producing an unexpectedly strong fitness defect. This is in particular the case of the T-lpoB guide as well as another guide targeting *lpoB* (T-lpoB2), which also produces an unexpected fitness defect. As a definitive proof that the phenotype produced by these two guide RNAs was not due to on-target activity, we deleted the *lpoB* gene in strain LC-E18. This deletion itself produced no growth defect, but the T-lpoB and T-lpoB2 guides still generated a strong fitness defect (Fig. 2d).

**Machine-learning approach reveals toxic seed sequences**. We then turned to a machine-learning approach to understand whether some sequence features could explain the unexpected fitness defect produced by these guides (Supplementary Fig. 3). We first used a regression tree to predict log2FC using target orientation and position as unique input features in order to locate all the regions where guides consistently produce a fitness defect, i.e., essential and fitness genes. We then analyzed guides targeting outside of these "important" regions. These guides are not expected to produce a fitness defect but ~8% of them show a log2FC < −3.5. We reasoned that some sequence patterns might be predictive of the toxicity of these guides. We used a locally connected neural network to predict the log2FC of guide RNAs using the one-hot-encoded sequence as the only input feature. A first model was trained using an arbitrary 60 nt region around the target and achieved a Pearson correlation coefficient of 0.54 (Root Mean Square Error (RMSE): 0.82) between the measured and predicted fitness on a held-out test set (Fig. 3a). The model thus seems to learn some features explaining the toxicity observed for guide RNAs such as T-lpoB.

To understand what information the model used to make predictions, we performed in silico experiments. We generated a set of 1000 random sequences and measured the effect on the model prediction of mutating each position along the sequence. This revealed that the model uses the whole 20 nt of the guide sequence, and in particular the 5 PAM-proximal bases, but not the surrounding region to make its predictions (Fig. 3b). We thus trained a second model using only the 20 nt of the target sequence. This model performed slightly better (Pearson-*r*: 0.56, RMSE: 0.81) and was used in the rest of the analysis.

When doing this in silico mutational analysis, one can also see that the effect of individual mutations depends a lot on the

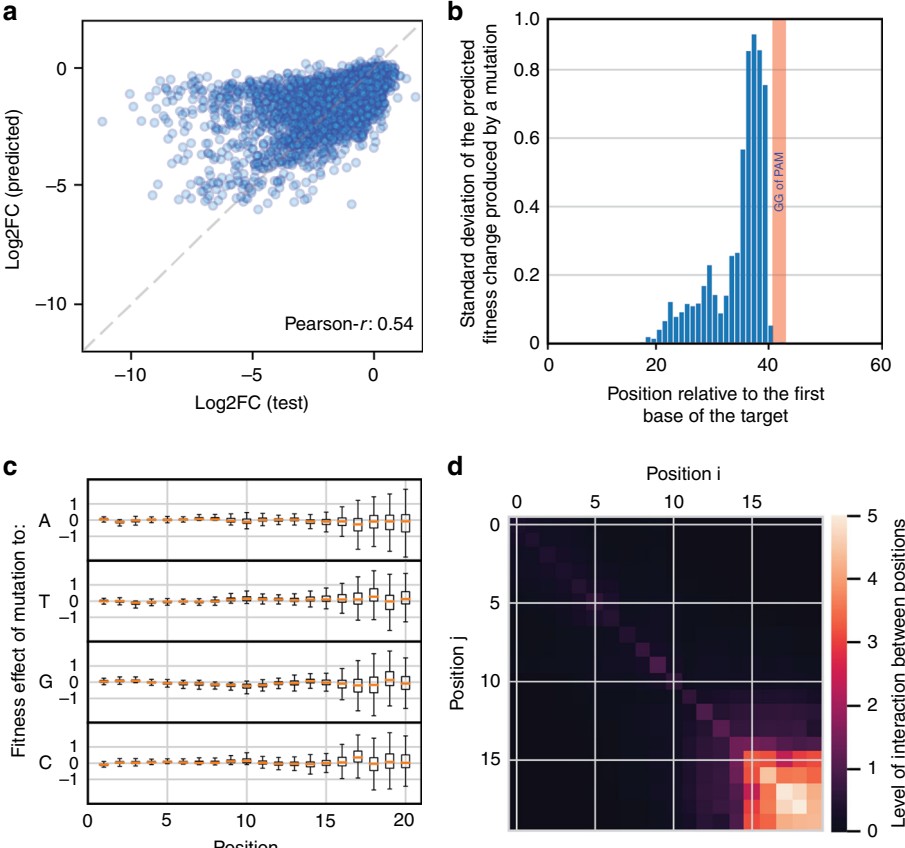

**Fig. 3** A machine-learning approach reveals a toxicity effect determined by the seed sequence. **a** A locally connected neural network was trained to predict the fitness effect of guide RNAs that target neutral regions, using the one-hot-encoded 60 nt sequence window around the target. Comparison of predicted and actual log2FC values on a held-out test set. **b** To identify the positions used by the model to make its predictions, we generated a set of 1000 random sequences, mutated each position in silico, and computed the effect of each mutation on the model prediction. The standard deviation of the effect of mutations at each position is plotted. The red bar indicates the position of the GG bases of the PAM. **c** The model was trained again using only the 20 nt of the guide sequence. The box plots show the distribution of the effects that mutations to all possible bases have on the model prediction. One can see that the effect of specific mutations can be either positive or negative, revealing a strong dependence on the rest of the sequence. **d** To measure the level of interaction between positions, we generated all possible pairs of mutations for each sequence in a set of 100 random sequences and compared the effect of individual mutations to that of pairs of mutations. Positions are interacting if the effect of a double mutation (Eij) is different from the sum of the effect of the single mutations (Ei + Ej). The heat map shows the average Euclidean distance between Eij and Ei + Ej for all pairs of positions (see Supplementary Fig. 5). Note the strong network of interacting bases in the 5 nt of the seed sequence

sequence context (Fig. 3c and Supplementary Fig. 4). For instance, a mutation at position 18 to a G can have a positive or a negative impact on the fitness depending on the rest of the sequence. This suggests important interactions between bases. To analyze these interactions, we mutated in silico every pair of bases and compared the effect of individual mutations to that of pairs of mutations. An interaction is observed if the effect of a pair of mutation is not simply the sum of the effects of individual mutations (Supplementary Fig. 5). The analysis revealed a strong network of interaction among the last 5 bases of the target/guide (Fig. 3d and Supplementary Fig. 6).

We observed that the distributions of fitness effects produced by guides with any given five base seed sequence are remarkably narrow. For instance, all guides with an ACCCA seed sequence produce a strong fitness defect regardless of their target position, while all guides with an ATACT seed sequence produce an intermediate fitness defect (Fig. 4a). The T-lpoB and T-lpoB2 guides have an AGTTT and TGGAA seed sequence, which show an average log2FC of −3.2 and −5.1, respectively. All in all, 130 out of the 1024 possible combinations of 5 nucleotides show a significantly reduced fitness compared to the mean (single sample

$t$-test, $p < 0.01$ after Bonferroni correction). We then refer to the fitness defect produced by these seed sequences as the "bad-seed" effect. The average fitness effect and standard deviation for all 5 nt seed sequences is given in Supplementary Data 3.

We experimentally validated the effect of three additional guides targeting the template strand of non-essential genes with a TGGAA or ACCCA seed sequence (T-garD, T-yhhX, T-ydeO). We also designed four guides with either the ACCCA or TGGAA seed sequences but where the 15 other nucleotides of the guide were randomized (R1-ACCCA, R2-ACCCA, R1-TGGAA, R2-TGGAA). Guide RNAs were cloned on the psgRNA plasmid, transformed in strain LC-E18, and plated on petri dishes containing aTc. All these guides produced a strong fitness defect, with the ACCCA seed sequence leading to a ~1000× reduction in platting efficiency and the TGGAA seed sequence leading to a small colony phenotype, while the control sgRNA with a TCTCG seed showed no visible phenotype (Fig. 4b). These results are consistent with an action guided by the seed sequence itself regardless of the rest of the guide sequence. As a matter of fact, guides truncated down to 10 bp are still able to kill E. coli (Supplementary Fig. 7).

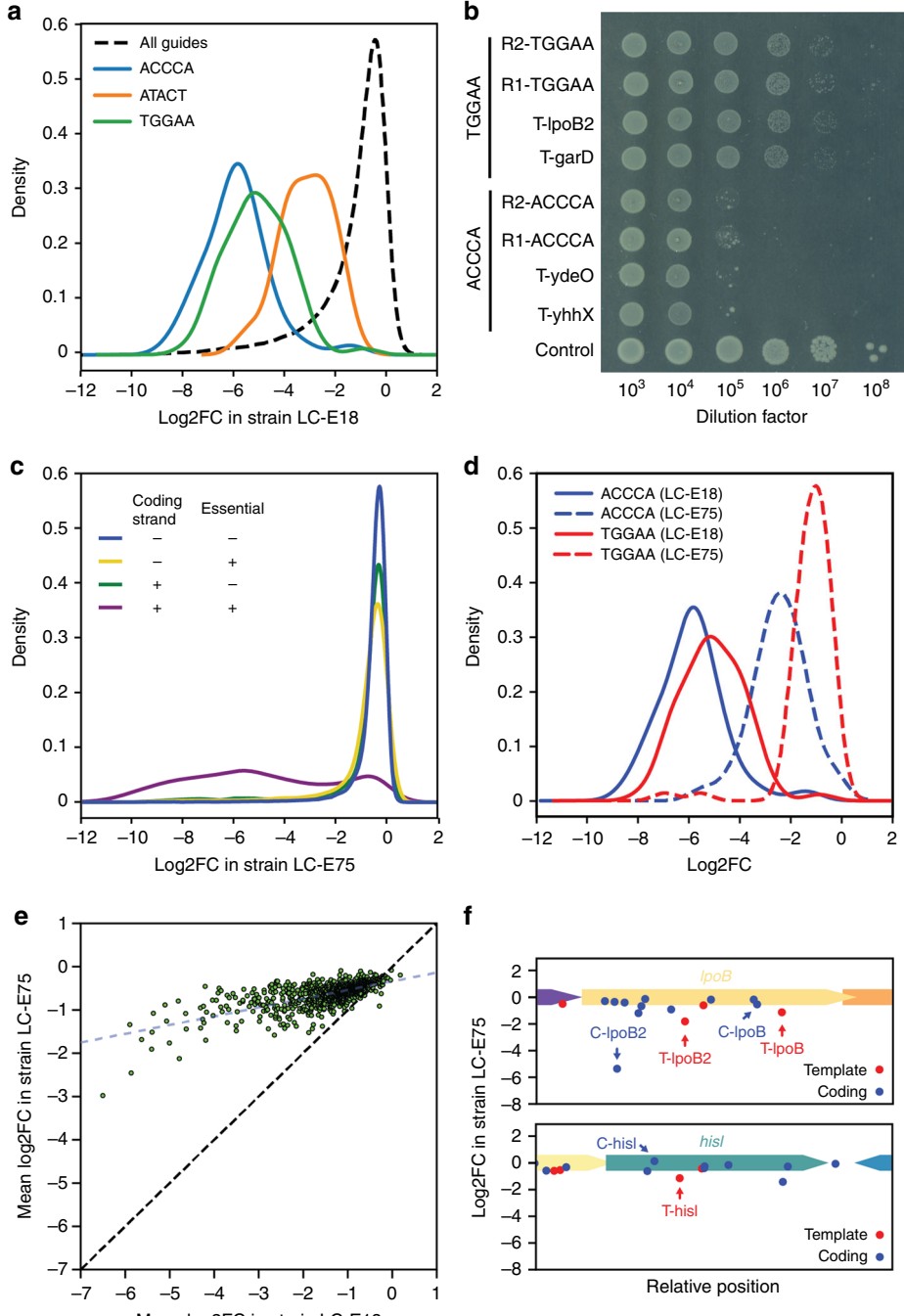

**Fig. 4** Specific 5 nt seed sequences produce strong fitness defects that can be alleviated by reducing dCas9 concentration. **a** Distribution of the fitness effect of guide RNAs that share specific 5 nt seed sequences as compared with the distribution of all the guides in the library targeting the template strand of genes. **b** Plasmid psgRNA was programmed with various guide RNAs sharing either the TGGAA seed sequence or the ACCCA seed sequence and introduced in strain LC-E18. Guides named R1 and R2 have sequences that do not match any position in the chromosome of *E. coli*. Cells were grown overnight, serially diluted, and plated on LB agar with 1 nM of aTc. Representative figure from four independent experiments. **c** Distribution of the fitness effect of guide RNAs in our library depending on target gene essentiality and target strand in strain LC-E75, which expresses dCas9 at a lower concentration than strain LC-E18. **d** Distribution of the fitness effect of guide RNAs sharing the ACCCA or TGGAA seed sequences in strain LC-E18 and LC-E75. **e** Average fitness effect of guides sharing specific 5 bp seed sequences in strain LC-E18 (*x* axis) and LC-E75 (*y* axis). The red line shows a linear regression (slope = 0.2, $R^2$ = 0.47). **f** Fitness effect of guides targeting gene *lpoB* and gene *hisI* in strain LC-E75. The strong fitness defect produced by C-lpoB2 can be explained by the presence of an off-target position in the *def* essential gene

**The bad-seed effect is alleviated at low dCas9 concentrations.** To better understand the mechanism of action of these "bad-seed" sequences, we selected mutants of the LC-E18 strain that could survive killing by the T-yhhX guide (ACCCA seed sequence) while maintaining an efficient repression of a target

*rpsL* gene. Six such mutants were obtained, and their genome sequenced. Unexpectedly, they all displayed mutations either in the promoter of dCas9 or frameshift mutations in dCas9 itself (Supplementary Table 2). Note that others have observed the same type of mutations in dCas9, suggesting that they are

relatively frequent[19]. The fact that these frameshift mutants still showed efficient *rpsL* repression indicates that they still express dCas9, likely through ribosome slippage, but do so at lower levels. This led us to hypothesize that the bad-seed effect is concentration dependent. A low dCas9 concentration is enough to block the expression of on-target positions, while a high dCas9 concentration is required to observe the bad-seed effect.

We thus designed a novel expression cassette that would reduce the expression level of dCas9 by introducing mutations in the RBS[20] (Supplementary Fig. 8a, b). We screened for the right level of expression by selecting clones that would result in cell death when the essential *rpsL* gene was targeted but that showed normal colony size in the presence of the R1-ACCCA or T-yhhC-ACCCA guide RNAs. The expression level of dCas9 was measured in several strains through western blot and correlated with the strength of the bad-seed effect (Supplementary Fig. 8c, d). The expression cassette selected in this manner displayed an expression level 2.6-time lower than the original strain LC-E18 and was integrated in strain LC-E75. It could repress a target *mCherry* reporter gene 91×, compared to the 167× repression obtained with the dCas9 expression cassette present in strain LC-E18 (Supplementary Fig. 9).

Strain LC-E75 carrying this fine-tuned Ptet-dCas9 cassette was then used to perform a genome-wide dCas9 knockdown screen following the same protocol as the screen previously performed with strain LC-E18. As expected, many guides that produced a strong fitness defect in strain LC-E18 had a weaker or no effect in strain LC-E75, but targets in the coding strand of essential genes still produced a strong fitness defect (Fig. 4c and Supplementary Fig. 10). When plotting the fitness effect of guides sharing a given seed sequence, one can see that the "bad-seed" effect is largely alleviated in strain LC-E75 but not abolished (Fig. 4d, e). Only 14 seed sequences still produced a moderate or weak significant effect (Supplementary Data 3). This new dCas9 expression cassette also makes the general quality of the screen better as the effect of targets within the same gene is now much more consistent (compare Fig. 4f and Fig. 2a; Supplementary Fig. 11). The number of guides producing a strong fitness defect (log2FC < −3.5) while targeting the template strand of non-essential genes dropped from 2499 to 532, due in the most part to the alleviation of the bad-seed effect.

The mechanism of the bad-seed effect described above remains to be elucidated. The binding of dCas9 with only 5 nt of identity between the seed and the target is likely too weak and transient to have a substantial effect on target gene expression. We verified this by targeting the promoter or open-reading frame of a mCherry reporter gene with only 5 nt of identity in the PAM-proximal region (Supplementary Fig. 12). At best, a 14% repression could be observed. Since blocking the expression of any single gene by only 14% is unlikely to stop the growth of *E. coli*, the bad-seed effect is likely due to dCas9 binding at many positions simultaneously or to an entirely different phenomenon. Note that the number of off-targets with a perfect identity of 5 nt in the PAM-proximal region and the fitness defect produced by bad-seed sequences are not correlated (Supplementary Fig. 13).

**A 9 nt match to the seed sequence can produce off-target effects**. The bad-seed effect is thus a different phenomenon than the off-target effect described above for guides like T-hisI, which block the expression of essential genes. We made the estimate that ~13% of the guides producing an unexpected fitness defects

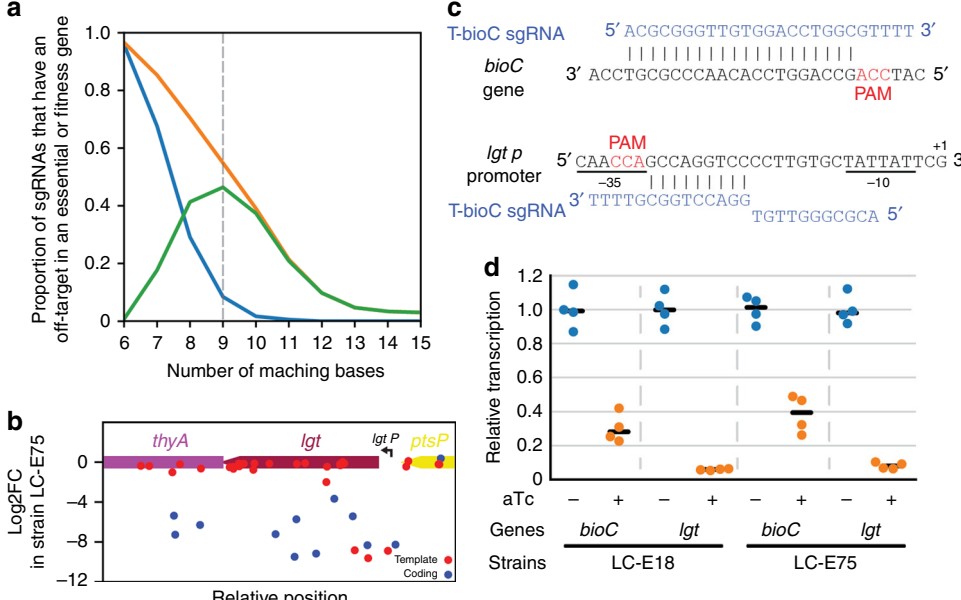

**Fig. 5** Off-targets with only 9 nt of identity to the seed sequence can produce strong fitness defects. **a** We plot here the proportion of guide RNAs that have an off-target position in a region where guides consistently produce a strong fitness defect. This proportion is shown for guides that target the template strand of non-essential genes but produce an unexpected fitness defect (orange), as well as for guides in the same genes and orientation but that do not produce a fitness defect (blue). This blue curve can be interpreted as the false-positive rate. The green curve is the difference between the green and blue curves. It can be interpreted as an estimate of the proportion of guides whose fitness defect is due to an off-target effect. The maximum is obtained for a perfect match of 9 nt in the seed sequence, which indicates that 9 nt of identity in the seed sequence is enough to produce a strong fitness defect, but mostly false-positive off-target positions are detected when going down to 8 nt of identity. **b** Fitness effect of guides targeting gene *lgt* in strain LC-E75. The strong fitness defect produced by the T-bioC guide can be explained by the presence of an off-target position in the promoter of essential gene *lgt*. **c** Off-target position of the T-bioC guide in the promoter of *lgt* essential gene. **d** Repression of *bioC* and *lgt* expression by the T-bioC guide in strains LC-E18 and LC-E75 as measured by RT-qPCR. Points show biological replicates (*n* = 4), the black bar shows the median

in strain LC-E18 have a likely off-target position to an essential or fitness gene, with 9 nt of identity or more to the seed sequence. In the screen performed in strain LC-E75, this number can now be estimated to be 45%. The same analysis was also performed while considering a minimum perfect match in the PAM-proximal region ranging from 6 to 15 bp (Fig. 5a and Supplementary Fig. 14). A seed size of length nine gave the largest difference between the positive detection rate (proportion of guides producing an unexpected fitness defect for which an off-target to an important region is detected) and false-positive detection rate (proportion of guides that do not produce an effect and for which an off-target to an important region is detected), suggesting that 9 nt of identity can be sufficient to block gene expression. We experimentally verified the fitness defect produced by one such guide, T-bioC, which has a candidate off-target position of 9 nt and a proper NGG PAM in the promoter of the essential gene *lgt* (Fig. 5b, c). This guide is indeed able to efficiently block the expression of *lgt* as measured by quantitative PCR (qPCR; Fig. 5d).

Note that, in strain LC-E75, the reduced dCas9 concentration not only alleviated the bad-seed effect but to a lesser extent also limited off-targeting. This can, for instance, be seen for the T-hisI target described above that showed a strong fitness defect in strain LC-E18 and no effect in the new screen (compare Fig. 2a and Fig. 4f). Conversely, the T-bioC off-target effect on *lgt* is still as strong in strain LC-E75 as in strain LC-E18 (Fig. 5d). Another example of a guide that still shows a strong off-target effect can be found at the beginning of the *lpoB* gene (C-lpoB2, Fig. 4f). This guide RNA has a perfect match of 14 bp between its seed sequence and a target on the coding strand of the essential *def* gene.

## Discussion

We performed here an unbiased screen of dCas9 effect on the growth of *E. coli*. Our results shed light on important design rules to consider when performing CRISPRi assays:

(i)    Targeting promoter regions in both orientations leads to strong silencing, but targeting the coding strand is required to block transcription elongation as described previously[6,7].

(ii)   Guides targeting genes in operons block the expression of all the downstream genes, while guides targeting downstream genes do not substantially affect the expression of the upstream genes.

(iii)  Guides that have off-targets in the genome with 9 nt of perfect identity or more to the seed sequence should be avoided. Unfortunately, such off-target positions are too frequent to be avoided easily. Guides that target the chromosome of *E. coli* MG1655 have a median of 4 off-targets that carry a perfect match of 9 nt or more with the seed sequence and a NGG PAM motif (see distribution in Supplementary Fig. 15). One can limit the chances that these off-targets will influence the phenotype under study by making sure that they fall in neutral regions, away from regulatory elements, and on the template strand of genes rather than on their coding strand, but researchers should be aware that this could be a confounding factor in their experiments.

(iv)  dCas9 concentration needs to be tuned to avoid the "bad-seed" effect while maintaining good on-target repression. Even under such conditions, it is preferable to avoid using guides that carry the strongest "bad seeds" identified here.

(v)   For the reasons described above, the effects of genes on a given phenotype should ideally not be inferred from the effect of a single guide but rather from the statistical analyses of several guides.

All in all, the results of the new screen performed with a reduced dCas9 concentration are much more consistent than the initial screen, but there remains a few hundred guide RNAs out of the 84,215 whose fitness effect could not be easily assigned to the bad-seed effect, a polar effect, or an off-target effect. Understanding the effect of these guides on fitness will require further analysis and might reveal interesting biology.

The dataset generated will also likely prove useful in future studies to decipher the determinants of dCas9 repression strength. Indeed, we observed some variability among guides targeting within the same essential gene in the same orientation, suggesting that some guides block expression better than others.

Perhaps the most puzzling question raised by this work is that of the mechanism responsible for the "bad-seed" effect. Others previously reported a dCas9 toxicity at high concentrations, which is likely the same as what we report here[21]. On average, we can find 280 positions in the chromosome of *E. coli* that can be bound by any 5 nt seed sequence with the requirement of a NGG PAM. It is unclear whether weak dCas9 binding at one, a few, or hundreds of positions simultaneously is required for this effect. The fact that it is observed at high dCas9 concentrations and that we were not able to identify mutants that can rescue cells by other means than reducing dCas9 expression suggests that binding to several positions is required. Our work also does not allow excluding the hypothesis that binding to other substrates than DNA might be responsible for the effect. While we further investigate this phenomenon, the results presented here are directly relevant to anyone using dCas9 in bacteria.

## Methods

**Bacterial strains and media**. *E. coli* strain MG1655 was obtained from the laboratory of Didier Mazel, Institut Pasteur. Cells were grown in Luria-Bertani (LB) broth. LB agar 1.5% was used as solid medium. Different antibiotics (20 μg/ml chloramphenicol, 100 μg/ml carbenicillin, 50 μg/ml kanamycin) were used as needed and lower concentrations were used to select for the integration of vectors in the chromosome (10 μg/ml chloramphenicol or 20 μg/ml kanamycin). *E. coli* strain DH5α (New England Biolabs) or MG1655 were used as transformation recipients for plasmid construction.

**Plasmid cloning and strain construction**. Linear vectors and inserts were generated by digestion with restriction enzymes or PCR and assembled through Gibson assembly[22] as detailed in Supplementary Table 3. Primers and plasmids are listed in Supplementary Tables 4, 5. Guide RNAs were cloned into plasmid psgRNA or psgRNAc by golden gate assembly[23]. The sequence of these plasmids is provided as supplementary Data 4. A list of all guides and corresponding primers is given in Supplementary Table 6.

The pOSIP plasmids[24] were used to integrate genetic elements at phage-attachment sites in the chromosome of *E. coli* K12 (MG1655). All the integrations were verified by PCR, and the backbones were flipped out using the pE-FLP plasmid[24]. Supplementary Table 7 summarizes the construction of strains LCE-18, LCE-75, and AV04.

Deletion of gene *lpoB* was performed using the lambda red recombineering strategy[25]. A linear DNA fragment was generated by PCR using pKD3 as a template with primers LC961 and LC962, followed by electroporation into MG1655 carrying plasmid pKOBEG-A[26]. Colonies resistant to chloramphenicol were selected and the resistance gene was removed using plasmid pE-FLP.

**Library construction**. The library was designed by randomly choosing targets with a proper NGG PAM around the genome of *E. coli* strain MG1655 (NC_000913.2). A pool of 92,919 oligonucleotides (synthesized by CustomArray) was amplified with primers LC296 and LC297 using the Phusion DNA polymerase (Thermo Scientific) over 18 cycles. The psgRNA backbone was PCR amplified using primers LC293 and LC294. Both the vector backbone and library insert were gel purified, followed by Gibson assembly. To avoid the introduction of bottlenecks in the library, a total of 19 transformation assays were performed each using 0.2 μl of Gibson assembly product and 20 μl of MG1655 electro-competent cells and plated on $12 \times 12$ cm$^2$ petri dishes resulting in a total of about $10^7$ colonies. Colonies were allowed to grow for only 4–5 h at 37 °C before pooling all the cells together. The plasmid library was then extracted from 5 ml of pooled colonies using a miniprep kit (Macherey-Nagel). The resulting plasmid library DNA was further transferred to strains LC-E18 and LC-E75 by electroporation.

**dCas9 knockdown assay**. Strain LC-E18 or LC-E75 containing the psgRNA library were grown from 1 ml aliquots frozen at −80 °C into 1000 ml LB until OD600 of 0.2. The expression of dCas9 was then induced by addition of aTc to a final concentration of 1 nM. Cells were grown for 17 generations by diluting the culture 100-fold once it reached OD600 2.2–2.5. This step was repeated twice. The plasmid library was extracted from 50 ml of culture at the beginning of the experiment (OD600 ~0.2) and 5 ml of the culture at the end of the experiment (OD600 ~2.2–2.5). The experiment was performed in triplicates starting from independent aliquots of the library generated from independent electroporation assays.

**Library sequencing**. A customized Illumina sequencing method was designed to avoid problems arising from low library diversity when sequencing PCR products. Two nested PCR reactions were used to generate the sequencing library with primers described in Supplementary Table 8. The first PCR adds the first index. The second PCR adds the second index and flow cells' attachment sequences. Sequencing is then performed using primer LC609 as a custom read 1 primer. Custom index primers were also used: LC499 reads index 1 and LC610 reads index 2. Sequencing was performed on a NextSeq 500 benchtop sequencer. The first 2 cycles that read bases common to all clusters were set as dark cycles, followed by 20 cycles to read the guide RNA. Using this strategy, we obtained on average 7.5 million and 17 million reads per experimental condition for LC-E18 and LC-E75, respectively.

**Fold-change computation**. The fold change in abundance of each guide RNA was computed from read counts using DESeq2[27] using data from the three replicates and normalized to the control guide 5′-TGAGACCAGTCTAGGGTCTCG-3′. Guides with a total number of reads across samples <20 were discarded from the analysis. A list of all targets with computed fold-change values is provided as Supplementary Data 5.

**Machine learning**. To model the fitness effect of guide RNAs, a regression tree was first fitted using only two features: the target orientation and position along the genome[28]. This allows the identification of all chromosomal regions that show a consistent fitness effect when targeted in a specific orientation (Supplementary Fig. 3). We were then interested in predicting the fitness effect of guide RNAs targeting regions where the regression tree predicts no substantial fitness defect (log2FC > −3.5).

The dataset was split into training, validation, and test sets. Several network architectures, including dense, sparse, and convolutional, were implemented using Keras and TensorFlow and hyperparameters were manually tuned on the validation set. The dataset used can be found in Supplementary Data 5 and the indices of the rows used for training, validation, and test sets can be found in Supplementary Data 6. The model used in this study consist in a sparsely connected network with 4 layers of size 40, 20, 10, and 5 where each neuron is only connected to the 5 proximal neurons in the previous layer. A tanh activation was used. The network was trained to minimize the mean square error of the log2FC prediction with L2 regularization using the Adam optimizer[29]. Training was interrupted when loss on the validation set ceased do decrease for more than two epochs. Note that we only performed a manual tuning of the architecture and hyperparameters of the model. Better models can likely be built, but this should not affect the conclusions of the study. The machine-learning approach performed here is summarized in Supplementary Figure 3, and the code is available as a jupyter notebook at the following address: https://gitlab.pasteur.fr/dbikard/badSeed_public.

**Reverse transcription-qPCR**. Overnight cultures were diluted 1:100 in 3 ml of LB, grown 1 h, and induced by addition of aTc. Cells were further grown 2 h followed by RNA extraction from 2 ml of culture using Trizol. All the RNA samples were treated with DNase (Roche) and reverse transcribed into cDNA using the Transcriptor First Strand cDNA Synthesis Kit (Roche). Dual-labeled probes (5'FAM, 3'BHQ1) were used to perform qPCR with the FastStart Essential DNA Probes master mix (Roche) in a LightCycle 96 (Roche). The 5'FAM is a reporter that is quenched by the 3'BHQ1 but released upon amplification of the target DNA. The FastStart Essential DNA Green Master Kit (Roche), which contains DNA Polymerase and double-stranded DNA-specific SYBR Green I dye, was used for the qPCR of bioC off-target effect. The fluorescence signal is directly proportional to the amount of target DNA. Primers and probes used are listed in Supplementary Table 9. Relative gene expression was computed using the ΔΔCq method[30].

**dCas9 induction on agar plates**. Strain LC-E18 or LC-E75 carrying specific psgRNA plasmids were grown overnight, followed by serial dilutions, and plating of 5 μl spots on LB agar plates with or without aTc (1 nM). Plates were incubated overnight and scanned using an Epson Perfection V550 Photo Color Scanner with a black background.

**Selection of mutants surviving the bad-seed effect**. An overnight culture of LC-E18 carrying plasmid psgRNAc::T-yhhX was plated on LB agar with 1 nM aTc and Kanamycin. After an overnight incubation at 37 °C, plates showed colonies of different sizes. Twenty-one big colonies were selected. The psgRNA plasmid carrying a guide RNA targeting the essential rpsL gene (C-rpsL) was then introduced

into these mutants. Clones where dCas9 is still able to efficiently block the expression of rpsL are expected to die. We selected 9/21 positive clones that were killed in this assay. As a final confirmation that these clones suppressed the bad-seed effect, we introduced a psgRNA plasmid carrying the R1-ACCCA guide. In all, 6/9 clones tolerated the expression of this sgRNA. The genomes of these 6 clones were sequenced to identify mutations (Supplementary Table 2). Genomic DNA was fragmented using a Covaris E220 ultrasonicator and sequencing libraries were prepared with the NEXTflex PCR-free DNA-Seq Kit (Bioo Scientific Corporation). Sequencing was done on a HiSeq2500 with paired-end reads of 100 bases.

**Fine-tuning of dCas9 expression level**. A library of RBS sequences was introduced in front of dCas9 to obtain a library of strains producing varying levels of dCas9. Primer LC1088 includes ambiguous positions leading to 64 different variants and was used to amplify and clone dCas9 on plasmid pOSIP-CO-RBS-library-dCas9 (Supplementary Table 3). The plasmid library was integrated into the chromosome of strain MG1655 and individual colonies were screened as follows: the psgRNA:rpsL plasmid was introduced into the cells to identify variants where dCas9 is still expressed at sufficient levels to efficiently block the expression of a target gene. The psgRNA:R1-ACCCA and psgRNA:T-yhhX plasmids were introduced to identify clones that would not produce the bad-seed effect. These plasmids carry a lambda cos site, which facilitated the screen of a large number of clones by enabling their transfer in the recipient strains via transduction with phage lambda as previously described[31]. Finally, 4 candidate colonies were selected from a total of 96 screened colonies. The pOSIP backbone was removed using pE-FLP, and the selected clones were named LC-E69, LC-E70, LC-E71, and LC-E72. Strain LC-E75 was constructed by integrating plasmid pIT5-KL-mcherry into strain LC-E69.

**Repression of mCherry by the fine-tuned dCas9 expression cassette**. The day before the measurement, cells were grown in 1 ml LB supplemented with 1 nM of aTc and 50 μg/ml of kanamycin at 37 °C using a 96 deep-well plate table-top shaker (Eppendorf). The day of the measurement, cells were diluted 250 times in fresh medium with aTc and kanamycin and grown for 1 h 45 min at 37 °C to reach the exponential phase. Cells were then fixed with 4% formaldehyde and washed with phosphate-buffered saline. A strain without mCherry was used to measure the auto-fluorescence background. Fluorescence of single cells was measured using a Miltenyi MACSquant flux cytometer. Ten thousand cells were measured per replicate. The average fluorescence of each sample was calculated by taking the mean of the single-cell fluorescence values and subtracting the mean fluorescence of the background.

**Western blot**. Western blot analysis was carried out as previously described[32]. In brief, cells were harvest after 2 h of dCas9 induction in 2× Laemmli sample buffer (with 5% of 2-mercaptoethanol). Samples were run in NuPAGE® Novex® Bis-Tris gels in reducing condition. Proteins were transferred to polyvinylidene difluoride membranes at 15 volts overnight. Rabbit monoclonal antibodies to SpCas9 (ab189380, Abcam, diluted 10,000-fold) and rabbit polyclonal antibody to RecA (ab63797, Abcam, diluted 3,000-fold) were used. Goat anti-rabbit horseradish peroxidase (ab6721, Abcam) was used as secondary antibody to visualize the protein with chemiluminescence (ECL). A Syngene G:box machine was used to acquire the images, which were subsequently analyzed with imageJ. Supplementary Figure 16 shows the uncropped western blot pictures.

**Code availability**. The code used for the machine-learning approach is available as Jupyter notebooks at the following address: https://gitlab.pasteur.fr/dbikard/badSeed_public.

**Data availability**. The screen results are provided as Supplementary Data 4. Other relevant data supporting the findings of the study are available in this article and its Supplementary Information files or from the corresponding author upon request.

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

## Acknowledgements

We are grateful to Oz Solomon and Zohar Yakhini for helpful discussions, to Christiane Bouchier from the Institut Pasteur sequencing Platform, to Charlotte Cockram and Aimee Wessel for providing experimental assistance, and to Jerome Wong Ng for his assistance with data analysis. *E. coli* mutants were sequenced by the Institut Pasteur Genomics Platform, a member of "France Génomique" consortium (ANR10-INBS-09-08). This work was supported by the European Research Council (ERC) under the Europe Union's Horizon 2020 research and innovation program (grant agreement No. [677823]); the French Government's Investissement d'Avenir program; Laboratoire d'Excellence 'Integrative Biology of Emerging Infectious Diseases' [ANR-10-LABX-62-IBEID]; and the Pasteur-Weizmann consortium and Pasteur-Roux Fellowship to L.C.

## Author contributions

L.C. and D.B. designed the study and wrote the manuscript. L.C. performed the experiments. A.V. performed the mcherry fluorescence measurements. D.B., F.R, V.K., H.V. and L.C. analyzed the data.

## Additional information

**Competing interests:** The authors declare no competing interests.

