## [Peer Review File · Nature Communications]

Reviewers' comments:

Reviewer #1 (Remarks to the Author):

The authors present a study on bacterial CRISPRi toxicity using a genome-wide pooled screen in *E. coli*. This phenomenon is known in the field but deeper investigation has been lacking. Thus, this study would be a valuable contribution to the broader adoption and utility of CRISPR/Cas9 technologies in bacteria. There are, however, a few important points which the authors could address to clarify and strengthen this work prior to publication.

Generally, these concerns are focused around balancing the focus on guide RNA sequences in the context of a "genome-wide" screen.

1- The utility of CRISPRi for identification of essential bacterial genes from the whole genome has not been previously established. However, the authors state that CRISPRi has "already been used to identify essential genes" in *B. subtilis* and *S. pneumonia* (refs. 4 and 5). These referenced studies use CRISPRi to study only to the set of putative essential genes identified by other methods such as transposons, and thus CRISPRi was not used to identify the essentials. This is a critical premise which readers should have in mind when reading the manuscript.

2- More discussion, including quantitative measures, on potential off-target guide sequences could be included. Currently, there is minimal and confusing discussion. For example, it is unclear what the authors are referring to (lines 146-152) when they say "58% of them have a candidate off-target". Which set of genes is meant by "them"? The 1.7% of guides targeting the non-essential genes with a strong fitness defect or the whole set? Such a discussion would be helpful in the beginning (lines 74-75) to gauge the magnitude of the bad seed problem. What about RNA folding? Are the bad seeds predicted to interact negatively with the guide RNA scaffold?

3- Further explanation on the rationale for machine learning parameters would be helpful for readers. For example, what is the relevance of supplying 60nt sequence window around a guide RNA's target? Furthermore, it seems odd to use the set of guide RNAs targeting the template strand of essential genes as a training set. Since these guides did not produce a phenotypically expected outcome (lines 48-50), these results suggest that the CRISPRi system is not functional. As such, the use of this dataset trains the discriminator with a broken system where the guides may be irrelevant, not functional guides in an active CRISPRi system.

4- Are there any positional enrichments to the bad seeds? A position weight matrix, or a weblogo visualization may be a helpful summary or depiction for readers.

5- To aid reproducibility and interpretation, the large supplementary data tables could be more adequately annotated. For example, it is not clear what the column labels in the files mean. It would be relevant to deposit raw files and any relevant computer processing code.

6- Aside from the bad seeds, an important takeaway from this study is to reduce the expression of dCas9 levels. The authors tantalizingly report a ~2x reduction in repression between their original dCas9 construct compared to their improved expression cassette (lines 134-136). It would be highly relevant if the authors could provide information on the reduction of dCas9 molecules for their isolated mutants with desired function, not only the magnitude of its functional outcome.

7- A third design rule emerges from this study, namely the use of multiple guides to summarize the effect of gene inhibition on cell fitness. As well shown by this study, single guides may be poor

indicators of that gene's fitness effect due to unintended consequences known or otherwise. However, a statistical summary of multiple guide RNAs, as depicted in Fig 1b,c,d, could yield a robust indication independent of outliers such as those with bad seeds. It would be relevant to include a discussion on this as it falls under design rules for CRISPRi screens.

Reviewer #2 (Remarks to the Author):

In this work the authors performed a genome-wide dead Cas9 (dCas9) knockdown screen in *E. coli* testing a library of ~84,000 unique guide RNAs. The main aim of the study consists in defining the properties and rules of such screens and estimate the off-target activity of dCas9. The statistical analysis of the experimental data, based on a machine learning approach, revealed that guide RNAs sharing specific 5 nucleotides can induce strong fitness defects independently from the rest of the sequence and the target position. This interesting result is important for the development of CRISPRi technique in bacteria

Although the authors show important experimental data, I believe that the quality of the analysis of the results presented in the manuscript needs to be improved. Below I reported few suggestions to make the paper acceptable for publication on Nature Communication.

Major revisions

1) The machine learning approaches used in this work are poorly described.

a) It seems there is a first method for fitting the fitness effect. What are the input features of the methods? What does it mean that "regression tree was first fitted using only the target orientation and position along the genome". Is the machine learning method taking only two input features? Please include a figure to show the architecture of the method and its input features.

b) In a second step, the authors aim to predict the fitness effect of guide RNAs targeting the template strand where regression tree predicts no substantial fitness defect. Also in this case, what are the input features of the neural network based approach? How the sequence information are provided in input?

2) The neural network-based method takes in input sequence information that encodes for the nucleotides in the different positions. In this case the usage of connections with only 5 proximal neurons can limit the ability of detecting non local interaction. In other words, the authors should test neural networks with higher number of proximal connections to check whether the non-local interactions ($|i-j| > 5$) are present.

3) To measure the level of interaction between the positions along the sequence the effect of single two single mutations (E_i, E_j) are compared with the effect of the double mutation (E_{ij}). Please define what is E and how it is calculated. Is it the fitness effect? Can you analyze the effect of mutation according to their types? I expect to see effects with different magnitude in case of single transition, transversions and double compensatory mutations.

4) The neural network algorithm has been tested dividing the whole dataset in training, validation and testing sets. The performance of the method can depend on the composition of the dataset. For this reason the splitting procedure for the selection of the training, validation and testing sets should be performed taking in consideration the redundancy of the dataset. I suggest to perform a more

stringent splitting procedure keeping in the same subset the sequences with high similarity.

5) The authors have tested the impact of dCas9 knockdown screen on a large set of genes. In the manuscript the cases of few selected genes (*lopB*, *hisI*, *ycfT*, *tyrS* etc) are extensively described. Is there any specific motivation for selecting those genes?

Minor Revision

1) The Figures 1a and 3c plot the distribution of the relative fitness for different genes and targets. Can the authors include in supplementary materials a tabular version of these plots reporting the percentage of cases falling within different ranges of relative fitness?

2) For the training, validation and testing of the machine learning methods specific subsets have been created. Please report as supplementary files the composition of those subsets.

3) Please rewrite in more appropriate format the equation used for the calculation of the relative fitness in the "Fitness computation" section.

4) It would be interesting to include in the manuscript a statistical analysis and the effect of relative fitness for guide RNA with low and high number off-target binding positions.

Reviewer #3 (Remarks to the Author):

Summary

Cui et al. performed a high-density CRISPR interference (CRISPRi) screen in *Escherichia coli* to better characterize the properties of such screens in bacteria. They make two key findings: 1) certain 5 nt seed sequences—i.e., "bad seeds"—in guide RNAs are toxic to *E. coli* regardless of the other 15 nt of guide sequence, and 2) 9-10 nt of base-pairing between the guide and target gene is sufficient to observe phenotypically meaningful repression in the context of a pooled fitness screen. The authors also reported that reduction of dCas9 levels had a mitigating effect on toxicity caused by bad seeds. These findings are critical for interpreting genome-scale CRISPRi libraries in bacteria, and are of great importance to the field of bacterial genetics. However, the manuscript would benefit from further description and analysis of the screening data.

Major points

1. The description of the guide RNA library used in this work is too vague to be a useful resource. The authors should clarify what they mean when they write that guide RNAs target "random positions." For instance, I assume that the guide targeted sequences were adjacent to PAMs in the genome, but that isn't indicated anywhere (that I found) in the text and methods. From what I could tell, there is no description of guide RNA design in the methods or elsewhere. I would guess that the authors made an effort upfront to identify guides with obvious off-target effects (e.g., guides that exactly target two locations in the genome), but there is no mention of this in the text or methods that I could find. Please describe the library design as completely as possible so that others can use it as a reference or template for their own CRISPRi screens.

2. The data analysis in this manuscript is mostly restricted to the bad seed issue, but there are several

other analyses the authors should perform to better characterize the behavior of CRISPRi in bacteria. Here are some points that I think deserve further analysis:

A) Polar effects. What are the effects of knocking down co-transcribed genes? Polarity on downstream genes is expected based on the mechanism of CRISPRi (Qi et al., 2013, Cell), but “reverse polarity” on upstream genes has also been observed in *Bacillus subtilis* (Peters et al., 2016, Cell). Do the authors observe forward and reverse polarity in their dataset? Polarity could likely be examined by analyzing operons with both essential and non-essential genes.

B) Patterns in guide RNA efficacy. Early studies (e.g., Qi et al., 2013, Cell) reported that CRISPRi is most efficacious when targeting the 5' ends of genes in bacteria, but it's unclear if that generalization would hold true in a larger data set. Is there any pattern in guide efficacy along the length of the gene? Further, the manuscript is focused on guides that have activity when they shouldn't (i.e., bad seeds), but are there guides that don't appear to be active that should be active (e.g., guides targeting the non-template strands of essential genes)? If these guides exist, are there any clues as to what would make the guide ineffective?

C. Behavior of intergenic guides. The authors have ~9000 guides that target areas outside of genes, but don't include any analysis of the efficacy (fitness phenotypes) of these guides. I assume that some of these guides target promoters of essential genes. How do the phenotypes of promoter targeted guides compare with those targeting the coding sequence? Do any of the intergenic guides give interesting phenotypes that can't be explained by the presence of a nearby gene or off-target/bad seed effect? Alternatively, are there sets of guides that do cause any measurable phenotypes and are therefore good candidates for control guide RNAs in future screens?

D. Essential genes. Did some genes considered essential by other methods (e.g., Keio collection) not show a fitness defect when targeted by CRISPR?

3. The authors suggest that reducing dCas9 level could mitigate the bad seed effect, but it's unclear how much dCas9 expression is needed to avoid toxicity. The authors should measure dCas9 protein levels in the cell to clarify this. Also, with a high-density CRISPRi screen such as the one performed here, isn't it possible to avoid bad seed issues by calculating fitness using the median guide?

4. I'm surprised that the relative fitness measurements for essential gene knockdowns center at around 0.8, rather than a much lower number. Could the authors please explain why the range of fitness values they obtained in a pooled experiment seems so narrow? It also appears to contrast with the very strong plating effects observed in Fig. 1E.

Minor points

1. There is a discrepancy in the number of generations of strain growth between the text (17 generations) and Fig. S1 (23 generations).

2. The neural network analysis should be described more clearly in the text, possibly by moving some information from the Fig. 2 legend and methods to the main text.

3. The axis labels on Fig. 2C are difficult to understand and should be substituted with something more descriptive.

4. Check for typos throughout—e.g., line 9 should read “transcription” instead of “transcriptions”, and line 23 should read “pneumoniae” instead of “pneumonia”.

5. Line 1; *Escherichia coli* should be spelled out the first time it's used.

6. Lines 22-23; these studies used CRISPRi to characterize essential genes phenotypes rather than to identify essential genes.

7. Line 34; Off-target activity for dCas9 in mammalian cells has been investigated (Gilbert et al., 2014, *Cell*) and should be cited here.

8. Line 121; Mutations in *dcas9* are expected and have been observed by others (Zhao et al., 2016, *J. Bacteriol.*).

9. Line 12; I think it's a little odd to say that bad seed kill *E. coli* "regardless of their target position," given that the actual mechanism is unknown. Maybe the authors could replace this with something like, "regardless of the other 15 nt of guide sequence"; this is closer to what was tested in the paper.

Reviewer #1 (Remarks to the Author):

The authors present a study on bacterial CRISPRi toxicity using a genome-wide pooled screen in *E. coli*. This phenomenon is known in the field but deeper investigation has been lacking. Thus, this study would be a valuable contribution to the broader adoption and utility of CRISPR/Cas9 technologies in bacteria. There are, however, a few important points which the authors could address to clarify and strengthen this work prior to publication.

Generally, these concerns are focused around balancing the focus on guide RNA sequences in the context of a “genome-wide” screen.

1- The utility of CRISPRi for identification of essential bacterial genes from the whole genome has not been previously established. However, the authors state that CRISPRi has “already been used to identify essential genes” in *B. subtilis* and *S. pneumonia* (refs. 4 and 5). These referenced studies use CRISPRi to study only the set of putative essential genes identified by other methods such as transposons, and thus CRISPRi was not used to identify the essentials. This is a critical premise which readers should have in mind when reading the manuscript.

This is indeed an important point. To avoid any confusion we have rephrased this sentence as follow:

“This provides a convenient method to silence genes that has already been used to investigate the role of essential genes in *Bacillus subtilis* and *Streptococcus pneumonia* via high throughput screens”

2- More discussion, including quantitative measures, on potential off-target guide sequences could be included. Currently, there is minimal and confusing discussion. For example, it is unclear what the authors are referring to (lines 146-152) when they say “58% of them have a candidate off-target”. Which set of genes is meant by “them”? The 1.7% of guides targeting the non-essential genes with a strong fitness defect or the whole set? Such a discussion would be helpful in the beginning (lines 74-75) to gauge the magnitude of the bad seed problem. What about RNA folding? Are the bad seed predicted to interact negatively with the guide RNA scaffold?

We have now modified the manuscript to discuss the off-targets more extensively in the place suggested by the reviewer, and give a more direct measure of the magnitude of the bad seed problem. You will find the modified section below, this comes after the description of the off-target of the T-hisI guide on the expression of the dnaKJ operon:

“This off-target position thus likely explains the fitness defect produced by T-hisI. To investigate the preponderance of guides with such off-targets, we looked for off-targets that displayed a perfect identity of 9nt or more between the seed sequence and regions where a strong fitness defect was consistently observed (i.e. essential or fitness genes). We found such off-targets for 24% (600/2499) of the guides that produced an unexpected fitness defect. As a control, we also looked at the proportion of guides with such off-targets among guides that target the same genes in the same orientation but produce no fitness defects. This occurs for 10.7% (3609/33612) of these guides, giving a measure of the false positive discovery rate. Guides that produce unexpected fitness defects are thus significantly more likely to have an off-target blocking the expression of a fitness or essential gene (Fisher exact test p -value <0.001). This enables to provide a conservative estimate that the fitness defect produced by 13% of these guides is due to their off-target activity.

When doing this analysis it became evident that no obvious off-target positions could be identified for a large majority of guide RNAs producing an unexpectedly strong fitness defect...”

Sequences that behave as bad seeds are very diverse, for instance AGGAA, TGACT and ACCCA are the strongest bad seeds but don't even have a single position in common. We couldn't find any association between the bad seed effect and potential secondary structure of the guide RNAs (scaffold included).

3- Further explanation on the rationale for machine learning parameters would be helpful for readers. For example, what is the relevance of supplying 60nt sequence window around a guide RNA's target? Furthermore, it seems odd to use the set of guide RNAs targeting the template strand of essential genes as a training set. Since these guides did not produce a phenotypically expected outcome (lines 48-50), these results suggest that the CRISPRi system is not functional. As such, the use of this dataset trains the discriminator with a broken system where the guides may be irrelevant, not functional guides in an active CRISPRi system.

The goal of the machine learning approach here is not to predict the efficiency with which guide RNAs can block transcription. We rather want to understand whether some sequence pattern can predict the unexpected toxicity produced by many guide RNAs that bind regions where they should not have any effect. We indeed want to make sense of a "broken system" here.

Using 60nt around the sequence window was an arbitrary choice to investigate whether regions surrounding the target could have an impact at all. We couldn't find any substantial impact and the final model which only uses the 20nt of the target performed better than models using the surrounding sequence. The machine learning approach is now described much more extensively in the results and methods section. It is also represented as a flow chart in supplementary figure 3.

4- Are there any positional enrichments to the bad seeds? A position weight matrix, or a weblogo visualization may be a helpful summary or depiction for readers.

There is no strong positional enrichment. While some patterns do come out of a weblogo representation, it is a very poor representation of the bad seed. Here is for instance a plogo computed on the 5nt seed sequence using (kplogo.wi.mit.edu):

While some bases do come out as more important than others, looking at this representation one would not be able to see that a seed like "CACTC" is one of the strongest bad seed. We thus prefer not to use this type of representation. We rather show the distribution of the effects of mutating *in silico* each position to specific bases in Figure 3b, as well as all possible twelve types of mutations in the newly added Supplementary Fig. 4 pasted below:

Supplementary Figure 4. Effect of mutations on the model predictions. We generated 1000 random sequences and computed the effect on the model prediction of mutating each base to all possible bases. The distribution of these effects are shown as boxplots for each type of mutation. The y-axis shows the difference in predicted log₂FC between the initial and the mutated sequence. The x-axis shows the position along the guide sequence. Transitions are highlighted in blue and transversions in red.

5- To aid reproducibility and interpretation, the large supplementary data tables could be more adequately annotated. For example, it is not clear what the column labels in the files mean. It would be relevant to deposit raw files and any relevant computer processing code.

We have now made all code and corresponding data tables available as jupyter notebooks at the following address, where the content of the tables is also extensively described:

https://gitlab.pasteur.fr/dbikard/badSeed_public

6- Aside from the bad seeds, an important takeaway from this study is to reduce the expression of dCas9 levels. The authors tantalizingly report a ~2x reduction in repression between their original dCas9 construct compared to their improved expression cassette (lines 134-136). It would be highly relevant if the authors could provide information on the reduction of dCas9 molecules for their isolated mutants with desired function, not only the magnitude of its functional outcome.

We have now performed a Western blot to quantify the reduction in dCas9 expression between the different constructions tested. These new results can be found in Supplementary Fig. 8 and the following sentence was added to the text:

“We thus designed a novel expression cassette that would reduce the expression level of dCas9, by introducing mutations in the RBS (Supplementary Fig. 8a, b). We screened for the right level of expression by selecting clones that would result in cell death when the essential rpsL gene was targeted, but that showed normal colony size in the presence of the R1-ACCCA or T-yhhC-ACCCA guide RNAs. The expression level of dCas9 was measured in several strains through Western blot and correlated with the strength of the bad seed effect (Supplementary Fig. 8c, d). The expression cassette selected in this manner displayed an expression level 2.6-time lower than the original strain LC-E18 and was integrated in strain LC-E75.”

7- A third design rule emerges from this study, namely the use of multiple guides to summarize the effect of gene inhibition on cell fitness. As well shown by this study, single guides may be poor indicators of that gene's fitness effect due to unintended consequences known or otherwise. However, a statistical summary of multiple guide RNAs, as depicted in Fig 1b,c,d, could yield a robust indication independent of outliers such as those with bad seeds. It would be relevant to include a discussion on this as it falls under design rules for CRISPRi screens.

Indeed, a statistical analysis of several guides in each gene can provide better estimates of the effect of a gene than looking at guides one by one. We have now rewritten the discussion to make a list of 5 design rules. We include this point as rule number 5:

“(v) For the reasons described above, the effects of genes on a given phenotype should ideally not be inferred from the effect of a single guide but rather from the statistical analyses of several guides.”

Reviewer #2 (Remarks to the Author):

In this work the authors performed a genome-wide dead Cas9 (dCas9) knockdown screen in E. coli testing a library of ~84,000 unique guide RNAs. The main aim of the study consists in defining the properties and rules of such screens and estimate the off-target activity of dCas9. The statistical analysis of the experimental data, based on a machine learning approach, revealed that guide RNAs sharing specific 5 nucleotides can induce strong fitness defects independently from the rest of the sequence and the target position. This interesting result is important for the development of CRISPRi technique in bacteria

Although the authors show important experimental data, I believe that the quality of the analysis of the results presented in the manuscript needs to be improved. Below I reported few suggestions to make the paper acceptable for publication on Nature Communication.

Major revisions

- 1) The machine learning approaches used in this work are poorly described.
 - a) It seems there is a first method for fitting the fitness effect. What are the input features of the methods? What does it mean that "regression tree was first fitted using only the target orientation and position along the genome". Is the machine learning method taking only two input features? Please include a figure to show the architecture of the method and its input features.

b) In a second step, the authors aim to predict the fitness effect of guide RNAs targeting the template strand where regression tree predicts no substantial fitness defect. Also in this case, what are the input features of the neural network based approach? How the sequence information are provided in input?

Indeed the regression tree only takes two input features and is simply used to identify regions and orientations in the genomes where guide RNAs have a consistent fitness effect. We then specifically investigate guides in regions where the regression tree does not predict a fitness defect, the goal being to understand why some guides in these regions are unexpectedly toxic. A neural network is trained on this data using the one-hot-encoded sequence as the only feature. We now described the machine learning approach much more extensively in the results and methods section. We also provide a schematic representation as supplementary figure 3, also pasted below. A jupyter notebook containing all the code necessary to reproduce the results is also made available at the following address:

https://gitlab.pasteur.fr/dbikard/badSeed_public

Supplementary Figure 3. Flow chart of machine learning analysis. A regression tree was first fitted using guide orientation and position as the unique features to predict log2FC. This enables to identify regions and orientations where guides consistently produce a fitness defect. The goal of this analysis was then to investigate the fitness defect produced by guides that target in “neutral” regions. In these regions, the log2FC of guides is not consistent and most guides have no effect. We kept guides in genomic regions where the prediction for the regression tree was greater than -3.5 and then fitted a neural network using the one-hot-encoded sequence as the unique feature to predict the log2FC. The neural network consists in 4 locally connected layers with a kernel size of 5 followed by a dense layer. More details are provided in the methods section.

2) The neural network-based method takes in input sequence information that encodes for the nucleotides in the different positions. In this case the usage of connections with only 5 proximal neurons can limit the ability of detecting non local interaction. In other words, the authors should test neural networks with higher number of proximal connections to check whether the non-local interactions ($|i-j|>5$) are present.

We initially started using dense layers that connect every position to each other, but saw strong improvements when using locally connected networks. The kernel size was manually tuned on a validation set. Note that all input positions are still connected to each other in the deeper layers and long distance interactions can thus still be encoded by our approach if relevant to the prediction. In order to ensure that the 5nt bad seed detected by our model is not a result of the 5nt kernel used in the locally connected layers, we also performed the same analyses with kernels of different sizes. See below the interactions plots obtained with a kernel size of 5 and 7, which look very similar:

3) To measure the level of interaction between the positions along the sequence the effect of single two single mutations (E_i, E_j) are compared with the effect of the double mutation (E_{ij}). Please define what is E and how it is calculated. Is it the fitness effect? Can you analyze the effect of mutation according to their types? I expect to see effects with different magnitude in case of single transition, transversions and double compensatory mutations.

E_i is the effect of introducing a mutation at position i. It is the difference between the predicted fitness of a given sequence and the predicted fitness of the same sequence but with a mutation at position i. Note that each position can be mutated to three different bases from the initial four possible bases, giving a total of 12 types of mutations. We now provide the plots of the effect of these 12 types of mutations in Supplementary Fig. X also pasted below. Transitions are highlighted in blue and transversions in red. While some patterns can be observed the main conclusion that can be drawn from this analysis is the importance of the seed sequence and the fact that the effect depends strongly on the sequence context as can be seen in the wide distribution of the effect of mutations in the seed sequence.

Supplementary Figure 4. Effect of mutations on the model predictions. We generated 1000 random sequences and computed the effect on the model prediction or mutating each base to all possible bases. The distribution of these effects are shown as boxplots for each type of mutation. The y-axis shows the difference in predicted log2FC between the initial and the mutated sequence. The x-axis shows the position along the guide sequence. Transitions are highlighted in blue and transversions in red.

When considering double mutants, one could consider $12 \times 12 = 144$ types of double mutations. We cannot easily provide these 144 plots here, but we provide the 16 plots corresponding to the effects of pairs of mutations towards the 4 possible bases (averaged over the possible bases of origin).

Supplementary Figure 6. Level of interaction predicted by the model for different combinations fo mutations. We generated all possible pairs of mutations for each sequence in a set of 100 random sequences, and compared the effect of individual mutations to that of pairs of mutations. The color shows the average Euclidian distance between the effect of a double mutation and the sum of the effect of single mutations (dark: no interaction, white: strong interaction).

4) The neural network algorithm has been tested dividing the whole dataset in training, validation and testing sets. The performance of the method can depend on the composition of the dataset. For this reason the splitting procedure for the selection of the training, validation and testing sets should be performed taking in consideration the redundancy of the dataset. I suggest to perform a

more stringent splitting procedure keeping in the same subset the sequences with high similarity.

Here we only use the sequence information to train the neural network, there are no obvious biases in the base pair composition of the target sequences, making it hard to split in a smarter way than random. We have now confirmed that our results still hold for different splits. We repeated the training procedure with 200 different random splits of the train+validation sets in the following proportion: train / validation / test => 80 / 10 / 10., keeping the test set untouched. The plot below shows the distribution of the pearson correlation coefficient on the test set that we obtained. The specific split used for the analysis performed in the paper is now provided in Supplementary Table 13.

5) The authors have tested the impact of dCas9 knockdown screen on a large set of genes. In the manuscript the cases of few selected genes (lopB, hisI, ycfT, tyrS etc) are extensively described. Is there any specific motivation for selecting those genes?

We initially plotted the effects of guides on ycfT and tyrS just as illustrations of the results of the screen, other genes could have been chosen. The new version of the manuscript now include more data about off-targets and polar effects and we thus decided to remove these two plots that were not really adding more information than figure 1a. Genes lpoB and hisI were chosen because of the presence of several guides showing unexpected effects, the choice of these genes rather than others was arbitrary. To make this clear we modified the manuscript as follow:

“We arbitrarily decided to further investigate two such guides targeting lpoB (T-lpoB) and hisI (T-hisI).”

Minor Revision

1) The Figures 1a and 3c plot the distribution of the relative fitness for different genes and targets. Can the authors include in supplementary materials a tabular version of these plots reporting

the percentage of case falling within different ranges of relative fitness?

A tabular version was added as supplementary table 3

2) For the training, validation and testing of the machine learning methods specific subsets have been created. Please report as supplementary files the composition of those subsets.

The composition of the subsets is now specified in supplementary table 13.

3) Please rewrite in more appropriate format the equation used for the calculation of the relative fitness in the "Fitness computation" section.

The fold changes in guide RNA abundance that we measure in this screen are not a measurement performed in steady state. In particular, one could expect essential genes to show a relative fitness much lower than the ~0.8 relative fitness that we reported on average, as pointed out by reviewer 3. This can be easily explained by several effects: first, after we start the experiment by adding aTc to the growth medium, it takes time for dCas9 to be expressed, bind its target and finally for the target protein to be diluted over the course of several divisions. As a result, cells can keep growing for several hours after we induce dCas9 to target an essential gene. Another important aspect is that our methods measure the relative abundance of plasmids carrying the guide RNA in the population. Plasmids inside dead cells will also be measured. As a result of these effects we still obtain reads at the end of the experiment for guide RNAs that actually kill E. coli as can be seen when we plate on agar with aTc. People typically use relative fitness to compare steady state growth of different genotypes. To avoid any confusion with the measurements performed here we have now decided to report $\log_2(\text{fold change})$ of the number reads obtained for each guide RNA, rather than relative fitness. The $\log_2\text{FC}$ are computed using data from 3 replicates with the DESeq2 software.

4) It would be interesting to include in the manuscript a statistical analysis and the effect of relative fitness for guide RNA with low and high number off-target binding positions.

The analysis performed in the manuscript shows that guides that produce an unexpected fitness defect are more likely to have at least one off-target to an essential or fitness gene compared to guides which do not produce a fitness defect. It makes sense that a single off-target blocking the expression of an essential gene is sufficient to produce a strong fitness defect. One might expect that having many off-targets could also generate pleiotropic effects that would be deleterious. This might for instance play a role in the bad seed effect.

We have now carried out further statistical analysis to investigate whether the number of off-targets to important regions or to the whole genome also matters. We build several linear models to predict fitness (\log_2 fold change in strain LC-E75) using as features:

- the presence or absence of at least one off-target in an important region with a perfect match in the seed sequence of 9nt or more (i.e. a region where guides consistently produce a fitness defect, as predicted by a regression tree)
- the number of off-targets in important regions with perfect matches in the seed sequence of length 5 to 10. (6 different features)
- the total number of off-targets in the genome with perfect matches in the seed sequence of length 5 to 10. (6 different features)

In all cases off-targets were only considered if they have a NGG PAM motif.

We used various model selection strategies including the leaps, stepAIC and pairwise ANOVA F-tests methods implemented in R. The only feature that was consistently selected is the presence or absence of off-targets (9nt of match with the seed) in important regions. More complex regression strategies including decision tree regression and gradient boosting were also attempted and yielded similar results. We conclude that the presence of at least one off-target with a seed match of 9nt to an important region is the only feature from this list that matters to predict the effect of a guide.

We have now added the following part to the results:

The mechanism of the bad seed effect described above remains to be elucidated. The binding of dCas9 with only 5nt of identity between the seed and the target is likely too weak and transient to have a substantial effect on target gene expression. We verified this by targeting the promoter or open reading frame of a mCherry reporter gene with only 5nt of identity in the PAM-proximal region (Supplementary Fig. 12). At best a 14% repression could be observed. Since blocking the expression of any single *E. coli* gene by only 14% is unlikely to stop the growth of *E. coli*, the bad seed effect is likely due to dCas9 binding at many positions simultaneously or to an entirely different phenomenon. Note that the number of off-targets with a perfect identity of 5nt in the PAM-proximal region and the fitness defect produced by bad seed sequences are not correlated (Supplementary Fig. 13).

Supplementary Figure 13. The bad seed effect does not correlate with the number of off-targets in the genome. Mean log₂FC for guides sharing the same 5nt seed sequence as a function of the number of off-targets in the genome of *E. coli* MG1655 that have a perfect match to these 5nt and a “NGG” PAM.

And

The bad seed effect is thus a different phenomenon than the off-target effect described above for guides like T-hisI which block the expression of essential genes. We previously made the estimate that ~13% (335) of the guides producing a strong fitness defects in strain LC-E18 while targeting the template strand of non-essential genes have a likely off-target position, with 9nt of identity to the seed sequence or more, in an essential or fitness gene. In the screen performed in strain LC-E75, this number can now be estimated to be 45% (239). The same analysis was also performed while considering a minimum perfect match in the PAM-proximal region ranging from 6 to 15bp (Fig. 5a and Supplementary Fig. 14). A seed size of length 9 gave the largest difference between the positive detection rate (proportion of guides producing an unexpected fitness defect for which an off-target to an important region is detected) and false positives detection rate (proportion of guides that do not produce an effect and for which an off-target to an important region is detected), suggesting that 9nt of identity can be sufficient to block gene expression.

Reviewer #3 (Remarks to the Author):

Summary

Cui et al. performed a high-density CRISPR interference (CRISPRi) screen in *Escherichia coli* to better characterize the properties of such screens in bacteria. They make two key findings: 1) certain 5 nt seed sequences—i.e., “bad seeds”—in guide RNAs are toxic to *E. coli* regardless of the other 15 nt of guide sequence, and 2) 9-10 nt of base-pairing between the guide and target gene is sufficient to observe phenotypically meaningful repression in the context of a pooled fitness screen. The authors also reported that reduction of dCas9 levels had a mitigating effect on toxicity caused by bad seeds. These findings are critical for interpreting genome-scale CRISPRi libraries in bacteria, and are of great importance to the field of bacterial genetics. However, the manuscript would benefit from further description and analysis of the screening data.

Major points

1. The description of the guide RNA library used in this work is too vague to be a useful resource. The authors should clarify what they mean when they write that guide RNAs target “random positions.” For instance, I assume that the guide targeted sequences were adjacent to PAMs in the genome, but that isn’t indicated anywhere (that I found) in the text and methods. From what I could tell, there is no description of guide RNA design in the methods or elsewhere. I would guess that the authors made an effort upfront to identify guides with obvious off-target effects (e.g., guides that exactly target two locations in the genome), but there is no mention of this in the text or methods that I could find. Please describe the library design as completely as possible so that others can use it as a reference or template for their own CRISPRi screens.

We indeed forgot to mention that the guides were chosen to target positions with a NGG PAM, sorry for the confusion. This important information was now added to the manuscript. Besides this constrain, guides were randomly picked among all possible guides targeting the genome. This random design was precisely chosen to perform an unbiased analysis of the behavior of dCas9 repression in *E. coli*.

2. The data analysis in this manuscript is mostly restricted to the bad seed issue, but there are several other analyses the authors should perform to better characterize the behavior of CRISPRi in bacteria. Here are some points that I think deserve further analysis:

We have performed all the analyses suggested by the reviewer below and were initially planning to include them in a separate manuscript. We have now decided to include the analyses corresponding the questions raised by the reviewer in points A, parts of point B and parts of point C in this study as a new section in the results and a new figure. We provide a detailed response to all points below.

A) Polar effects. What are the effects of knocking down co-transcribed genes? Polarity on downstream genes is expected based on the mechanism of CRISPRi (Qi et al., 2013, Cell), but “reverse polarity” on upstream genes has also been observed in *Bacillus subtilis* (Peters et al., 2016, Cell). Do the authors observe forward and reverse polarity in their dataset? Polarity could likely be examined by analyzing operons with both essential and non-essential genes.

B) Patterns in guide RNA efficacy. Early studies (e.g., Qi et al., 2013, Cell) reported that CRISPRi is most efficacious when targeting the 5' ends of genes in bacteria, but it's unclear if that generalization would hold true in a larger data set. Is there any pattern in guide efficacy along the length of the gene?

A new section and figure were added to address these points and are pasted below:

“In order to investigate the properties of dCas9 repression in *E. coli* we can analyze the effect of guides targeting essential genes. We expect guides that efficiently block the expression of these genes to be depleted from the library. Previous reports suggested that dCas9 efficiently blocks transcription elongation only when binding the coding strand (non-template strand)^{6,7}. As expected guide RNAs targeting essential genes have on average a strong fitness effect when they bind to the coding strand, and no fitness effect when they bind to the template strand (Fig. 1a). On the other hand, dCas9 binding in both orientations was reported to efficiently block the initiation of transcription. We analyzed the effect of guides binding the promoter region of a subset of 64 essential genes whose promoter is well defined (Supplementary Table 1). Our results also corroborate these findings (Fig. 1b).

Since dCas9 blocks transcription, we expect that targeting a gene in an operon will also silence all the downstream genes. Guides targeting non-essential genes upstream of essential genes in operons indeed showed a strong fitness defect (see the examples the *cydDC* and *ycaR-kdsB* operons in Fig. 1c). A reverse-polar effect was reported in *B. subtilis* where targeting downstream of a gene was seen to block the expression of the upstream gene likely through destabilization of the interrupted transcript¹⁸. In our screen we can find many examples where targeting a non-essential gene downstream of an essential gene does not have an impact on the cell fitness (see the examples of the *rpoZ-spoT-trmH-recG* and *psd-mscM* operons in Fig. 1d). Opposite examples where targeting the non-essential gene does have an effect can also be found, but in these cases the non-essential gene is typically known to be required for normal growth or is itself followed by another essential gene. These observations suggest that translation can still efficiently occur from mRNAs interrupted by dCas9. We did nonetheless observe that guides targeting within ~100nt after the stop codon of an essential gene sometimes produced a fitness defect. This can for instance be seen for a guide in Fig. 1c targeting just after *kdsB*. To study this in a more systematic way we compiled a list of essential or fitness genes which are not followed by another essential or fitness gene (Supplementary Table 2). Guides targeting within 100nt of the end of these genes on the coding strand indeed produce a weak but significant fitness defect (Fig. 1e, single sample t-test comparison to the mean log₂FC of guides targeting the template strand of genes, $p\text{-value} < 10^{-4}$). On the other hand guides targeting 100nt to 200nt after the end of these genes did not show a significant effect. A reverse polar effect of dCas9 on the expression of upstream genes thus does seem to exist in *E. coli*, but it is short range and weak.

Previous reports suggested that dCas9-mediated repression is negatively correlated with the distance from the beginning of the gene⁶. The same list of genes was used to look at the effect of the relative distance along the gene (Fig. 1f). No effect could be seen: dCas9 efficiency does not seem to correlate with the position inside the gene.”

Please also find the new figure 1 pasted below.

Figure 1. Effect of dCas9 binding position and orientation

(a) Distribution of the fitness effect of guide RNAs in our library depending on target gene essentiality and target strand. **(b)** Rolling average of the fitness effect produced by guides targeting the promoter of essential genes in both orientations (rolling window size of 50bp). **(c)** Examples of polar effect seen in the *cydDC* and *ycaR-kdsB* operons. Gene *cydC* and *kdsB* highlighted in red are essential but not gene *cydD* and *ycaR*. **(d)** Example of operons containing an essential gene followed by a non-essential gene. Targeting the downstream non-essential gene usually does not produce a fitness defect.

(e) Rolling average of the fitness effect produced by guides targeting the end of essential genes (rolling window size of 50bp). (f) Rolling average of the fitness effect produced by guides along the length of essential genes. Gene start is 0 and gene end is 1, (rolling window size is 5% of the gene length). In all rolling average plots the shaded area represents the standard deviation.

Further, the manuscript is focused on guides that have activity when they shouldn't (i.e., bad seeds), but are there guides that don't appear to be active that should be active (e.g., guides targeting the non-template strands of essential genes)? If these guides exist, are there any clues as to what would make the guide ineffective?

We can indeed find guides that don't appear to be active that should be active. In general, guides targeting nearby positions in a given essential gene can show different fitness defects. There is no straight forward explanation to this variability, but we were able to build a neural network model that can predict a small part of this variability and provides cues as to what makes some guides more efficient than others. This model, its experimental validation and the design of a novel library based on the model predictions will be the topic of a separate manuscript under preparation. We have added a sentence in the discussion to present this new research direction opened by the present study.

“The dataset generated will also likely prove useful in future studies to decipher the determinants of dCas9 repression strength. Indeed, we observed some variability among guides targeting a given essential gene in a given orientation, suggesting that some guides block expression better than others.”

C. Behavior of intergenic guides. The authors have ~9000 guides that target areas outside of genes, but don't include any analysis of the efficacy (fitness phenotypes) of these guides. I assume that some of these guides target promoters of essential genes. How do the phenotypes of promoter targeted guides compare with those targeting the coding sequence? Do any of the intergenic guides give interesting phenotypes that can't be explained by the presence of a nearby gene or off-target/bad seed effect? Alternatively, are there sets of guides that do cause any measurable phenotypes and are therefore good candidates for control guide RNAs in future screens?

The guides that fall in intergenic regions and show an effect are almost exclusively in the promoter region of essential genes or immediately after them. These guides are now used to perform the analysis detailed above as a reply to point A. Besides these guides, there remains several guides with unexpected strong fitness effects that cannot be easily explained by a polar effect, the bad seed effect or by an off-target position. Further work will be required to shed light on the effect produced by these guides. This point is mentioned in the discussion as follow:

“All in all, the results of the new screen performed with a reduced dCas9 concentration are much more consistent than the initial screen, but there remains a few hundred guide RNAs out of 84215 whose fitness effect could not be easily assigned to the bad seed effect, a polar effect or an off-target effect. Understanding the effect of these guides on fitness will require further analysis and might reveal interesting biology.”

Regarding the use of guides as controls, we think it is a better idea to use sequences that do not target any position in the genome (and avoid bad-seed sequences as well as off-targets) rather than guides that target neutral regions. Indeed, depending on the phenotypes that one might want to investigate we can never be sure that a position is really neutral. Here we only looked at fitness in rich medium and regions that appear neutral to us might not be neutral in other experimental conditions. We are currently working on a novel library design that includes such a list of control guides.

D. Essential genes. Did some genes considered essential by other methods (e.g., Keio collection) not show a fitness defect when targeted by CRISPR?

A detailed analysis of what our screen can teach us about gene essentiality in *E.coli* will be part of a separate study. We indeed find 65 genes in our analysis that do not show a fitness defect but are annotated as essential in the Ecogene database (<http://www.ecogene.org>). Out of them, 4 genes are part of toxin-antitoxin systems which are encoded in operons with the antitoxin gene preceding the toxin gene. Antitoxins are annotated as essential since deleting them is lethal. However, in our case, dCas9 blocks the expression of both the antitoxin and the toxin gene leading to no adverse effect on the cell fitness. In 14/65 cases, gene essentiality has been challenged by conflicting results and these genes are likely not essential under our experimental conditions. For the remaining 45 genes a simple explanation is that dCas9 doesn't completely block the expression of target genes but some protein production can remain. In some cases where the expression of a target gene is under the control of a feedback loop, blocking the expression of a protein will lead to an increased transcription rate from the promoter and an effective repression rate by dCas9 that can be quite low. We have indeed experimentally validated that this phenomenon occurs in a synthetic feedback loop. These results are published in a manuscript in which we actually propose to use this phenomenon to quantitatively measure genetic feedback: <https://www.biorxiv.org/content/early/2017/07/18/164384>.

3. The authors suggest that reducing dCas9 level could mitigate the bad seed effect, but it's unclear how much dCas9 expression is needed to avoid toxicity. The authors should measure dCas9 protein levels in the cell to clarify this. Also, with a high-density CRISPRi screen such as the one performed here, isn't it possible to avoid bad seed issues by calculating fitness using the median guide?

We have now performed a Western blot to quantify the reduction in dCas9 expression between the different constructions tested. This new results can be found in supplementary fig. 8 and the following sentence was added to the text:

"We screened for the right level of expression by selecting clones that would result in cell death when the essential *rpsL* gene was targeted, but that showed normal colony size in the presence of the R1-ACCCA or T-yhhC-ACCCA guide RNAs. The expression level of dCas9 was measured in several strains through Western blot and correlated with the strength of the bad seed effect (Supplementary Fig. 8c, d). The expression cassette selected in this manner displayed an expression level 2.6-time lower than the original strain LC-E18 and was integrated in strain LC-E75."

Supplementary Figure 8. The bad seed effect can be alleviated by reducing dCas9 concentration. A library of RBS controlling the expression of dCas9 was generated and clones were selected for their ability to survive the dCas9 expression in the presence of a bad seed sequence while still efficiently blocking the expression of *rpsL* when guided by the C-*rpsL* sgRNA. Four clones were selected (LC-E70, LC-E71, LC-E72 and LC-E75). (a) RBS sequence of the selected clones. (b) Cells carrying the T-yhhX, R1-ACCCA or C-*rpsL* guide RNAs were grown overnight, followed by serial dilution and plating with aTc. (c) Western blot image of selected dCas9 strains. (d) quantification results of 3 western blot (Uncropped photos are shown in Supplementary Figure 15).

It is indeed possible to use the median guide, or other statistical methods (as echoed by reviewer 2), to compute the fitness of a gene in order to avoid issues coming from inconsistencies between guide RNAs. We have now rewritten the discussion to make a list of 5 design rules which includes the following point:

“(v) For the reasons described above, effects of genes should ideally not be inferred from the effect of a single guide but rather from the statistical analyses of several guides.”

4. I'm surprised that the relative fitness measurements for essential gene knockdowns center at around

0.8, rather than a much lower number. Could the authors please explain why the range of fitness values they obtained in a pooled experiment seems so narrow? It also appears to contrast with the very strong plating effects observed in Fig. 1E.

One would indeed expect essential genes to show a relative fitness much lower than the ~0.8 relative fitness that we report on average. This can be easily explained by several effects: first, after we start the experiment by adding aTc to the growth medium, it takes time for dCas9 to be expressed, bind its target and finally for the target protein to be diluted over the course of several divisions. As a result, cells can keep growing for several hours after we induce dCas9 to target an essential gene. Another important aspect is that our methods measure the relative abundance of plasmids carrying the guide RNA in the population. Plasmids inside dead cells will also be measured. As a result of these effects we still obtain reads at the end of the experiment for guide RNAs that actually kill E. coli as can be seen when we plate on agar with aTc. People typically use the concept of relative fitness to compare steady state growth of different genotypes. Since our measurements are not performed in steady state and to avoid any confusion we have now decided to report $\log_2(\text{fold change})$ rather than relative fitness.

Minor points

1. There is a discrepancy in the number of generations of strain growth between the text (17 generations) and Fig. S1 (23 generations).

Thank you for picking this up. It is now corrected.

2. The neural network analysis should be described more clearly in the text, possibly by moving some information from the Fig. 2 legend and methods to the main text.

We now describe the machine learning methods more extensively and provide a schema of the analysis performed as supplementary figure 3 (also pasted in reply to reviewer 1 above). We also provide a jupyter notebook that recapitulates the whole analysis performed in the manuscript and is available at the following address:

3. The axis labels on Fig. 2C are difficult to understand and should be substituted with something more descriptive.

We have now updated the labels to make them easier to understand.

4. Check for typos throughout—e.g., line 9 should read “transcription” instead of “transcriptions”, and line 23 should read “pneumoniae” instead of “pneumonia”.

Thanks. Done.

5. Line 1; Escherichia coli should be spelled out the first time it's used.

Thanks. Done.

6. Lines 22-23; these studies used CRISPRi to characterize essential genes phenotypes rather than to identify essential genes.

This sentence was rephrased as follows:

“This provides a convenient method to silence genes that has already been used to investigate the role of essential genes in *Bacillus subtilis* and *Streptococcus pneumoniae* via high throughput screens.”

7. Line 34; Off-target activity for dCas9 in mammalian cells has been investigated (Gilbert et al., 2014, Cell) and should be cited here.

We have now modified the introduction to include a discussion of this reference:

“The action of Cas9 at off-target positions is a major concern for genome editing applications^{10–12}, as it could lead to undesired mutations. While extensive binding beyond the seed sequence is required for a conformational shift in Cas9 to occur leading to DNA cleavage¹³, chromatin immunoprecipitation sequencing (ChIP-seq) experiments have revealed that Cas9 can bind to target positions with as little as 5nt of homologies between the seed region of the guide RNA and the target^{14,15}, possibly binding hundreds of positions in genomes. These results are also consistent with *in vitro* assays showing that dCas9 binding to its target remains unaffected by up to 12 mismatches in the PAM-distal region¹⁶, as well as evidence that DNA binding guided by as little as 10 bases can be sufficient for dCas9 to have an effect on transcription in *E. coli*⁷. These results are however in sharp contrast to what was reported in a study of dCas9 mediated repression or activation in human cells, where activity was highly sensitive to mismatches¹⁷. While substantial work has already been conducted to characterize the off-target activity of Cas9 in Eukaryotes for genome editing applications, comparatively little has been done for dCas9 in general and in bacteria in particular. ”

8. Line 121; Mutations in *dcas9* are expected and have been observed by others (Zhao et al., 2016, J. Bacteriol.).

Thank you for pointing out this reference to us. Indeed, Zhao et al. observed the same type of mutation in dCas9 as us. We now cite them in the following sentence:

“To better understand the mechanism of action of these “bad seed” sequences, we selected mutants of the LC-E18 strain that could survive killing by the T-yhhX guide (ACCCA seed sequence), while maintaining an efficient repression of a target *rpsL* gene. Six such mutants were obtained and their genome sequenced. Unexpectedly, they all displayed mutations either in the promoter of dCas9 or frameshift mutations in dCas9 itself (Supplementary Table 5). Note that others have observed the same type of mutations in dCas9, suggesting that they are relatively frequent¹⁹. The fact that these frameshift mutants still showed efficient *rpsL* repression indicates that they still express dCas9, likely through ribosome slippage, but do so at lower levels.”

9. Line 12; I think it's a little odd to say that bad seed kill *E. coli* “regardless of their target position,” given that the actual mechanism is unknown. Maybe the authors could replace this with something like, “regardless of the other 15 nt of guide sequence”; this is closer to what was tested in the paper.

We agree with the reviewer and this was corrected as suggested.

REVIEWERS' COMMENTS:

Reviewer #1 (Remarks to the Author):

This manuscript is much improved and seems to adequately address my critiques (and others).

Reviewer #2 (Remarks to the Author):

In the new version of the manuscript the authors addressed almost all the issues raised in my previous review.

I believe that the manuscript can be accepted for publication after addressing the minor points reported below.

Minor Revision

1) In the Supplementary Dataset 13 are included 85,392 data records but only 70,308 of them are used for generating the Supplementary Table 3. Are the 15,084 data records not mapped to gene and without any associated sequence important? If not please remove them from the Supplementary Dataset 13.

2) In the calculation of the performance of the machine learning algorithm the authors reported the values of the Pearson's correlation coefficient as unique score of the performance. Can you include the value of the Root Mean Square Error?

3) The performance of the machine learning method can be scored in binary classification mode considering a $\log_2FC = -3.5$ as a threshold. In this case what will be the overall accuracy and Matthews Correlation Coefficient of the method?

4) Although there is no evidence for sequence bias in the training and testing step of the machine learning method, I would suggest to group the data by the associated gene names and randomly split them at gene level. This procedure is used to keep all the data corresponding to each gene either in the training or testing sets. Please verify that the gene-based splitting procedure does not significantly change the performance of the machine learning method.

Reviewer #3 (Remarks to the Author):

The authors fully addressed my concerns. I now feel this paper will be a major contribution to the field of bacterial genetics.

1) In the Supplementary Dataset 13 are included 85,392 data records but only 70,308 of them are used for generating the Supplementary Table 3. Are the 15,084 data records not mapped to gene and without any associated sequence important? If not please remove them from the Supplementary Dataset 13.

We initially designed a guide RNA library containing 92,919 guides. We obtained sufficient number of sequencing reads to compute reliable fold change values for 78,137 guides targeting a total of 85,392 positions in the genome (the number of lines in the table mentioned by the reviewer). Supplementary table 3 only contains guide targeting inside ORFs and with a unique target in the genome. The total number of these guides is indeed 70,308, guides in intergenic regions are used in the analyses performed for figure 1 and should not be discarded from the table. We now provide a more detailed explanation of this in the Methods section.

2) In the calculation of the performance of the machine learning algorithm the authors reported the values of the Pearson's correlation coefficient as unique score of the performance. Can you include the value of the Root Mean Square Error?

We have now included the RMSE values in the main text as well as a scatter plot of predictions vs. test values as a new panel in Fig. 3.

(60 bp model)

pearson-r: 0.54
spearman-r: 0.52
Root Mean Square Error: 0.82

(20 bp model)

pearson-r: 0.56
spearman-r: 0.55
Root Mean Square Error: 0.81

3) The performance of the machine learning method can be scored in binary classification mode considering a $\log_2FC = -3.5$ as a threshold. In this case what will be the overall accuracy and Matthews Correlation Coefficient of the method?

We have now also trained the model in binary classification mode as suggested by the reviewer, but using a threshold of $\log_2FC = -2$ to avoid having classes that are too biased. The neural network structure was kept identical to the one used in the regression model. We simply replaced the final activation function for a sigmoid, and trained the model with binary cross-entropy as a loss function. The model trained in this way obtained an accuracy of 0.86, a Matthews Correlation Coefficient value of 0.45. We also provide below a ROC analysis (AUC: 0.78). Since the problem we are modelling is typically a regression problem we do not see much value in adding these results to the manuscript.

4) Although there is no evidence for sequence bias in the training and testing step of the machine learning method, I would suggest to group the data by the associated gene names and randomly split them at gene level. This procedure is used to keep all the data corresponding to each gene either in the training or testing sets.

Please verify that the gene-based splitting procedure does not significantly change the performance of the machine learning method.

We have now retrained the 20bp model with a random split by gene as follow:

	Number of genes	Number of guides
Train set	3382	53793
Validation set	423	7086
Test set	423	6998

The results obtained were on par with the random splitting procedure. Note that the number of guides used for training was slightly smaller here as this splitting procedure does not allow including guides outside of genes:

pearson-r 0.54
spearman-r 0.55
RMSE: 0.83

These results are consistent with the fact that the bad seed effect was shown experimental to be independent of the target gene.